# The metabolome of Mexican cavefish shows a convergent signature highlighting sugar, antioxidant, and Ageing-Related metabolites

J Kyle Medley[1]*[†], Jenna Persons[1†‡], Tathagata Biswas[1], Luke Olsen[1,2], Robert Peuß[2§], Jaya Krishnan[1], Shaolei Xiong[1], Nicolas Rohner[1,2]*

[1]Stowers Institute for Medical Research, Kansas City, United States; [2]Department of Molecular and Integrative Physiology, University of Kansas Medical Center, Kansas City, United States

**\*For correspondence:**
medjk@comcast.net (JKM);
nro@stowers.org (NR)

[†]These authors contributed equally to this work

**Present address:** [‡]Newman University, Wichita, United States; [§]University of Münster, Münster, Germany

**Competing interest:** The authors declare that no competing interests exist.

**Abstract** Insights from organisms, which have evolved natural strategies for promoting survivability under extreme environmental pressures, may help guide future research into novel approaches for enhancing human longevity. The cave-adapted Mexican tetra, *Astyanax mexicanus*, has attracted interest as a model system for *metabolic resilience*, a term we use to denote the property of maintaining health and longevity under conditions that would be highly deleterious in other organisms (Figure 1). Cave-dwelling populations of Mexican tetra exhibit elevated blood glucose, insulin resistance and hypertrophic visceral adipocytes compared to surface-dwelling counterparts. However, cavefish appear to avoid pathologies typically associated with these conditions, such as accumulation of advanced-glycation-end-products (AGEs) and chronic tissue inflammation. The metabolic strategies underlying the resilience properties of *A. mexicanus* cavefish, and how they relate to environmental challenges of the cave environment, are poorly understood. Here, we provide an untargeted metabolomics study of long- and short-term fasting in two *A. mexicanus* cave populations and one surface population. We find that, although the metabolome of cavefish bears many similarities with pathological conditions such as metabolic syndrome, cavefish also exhibit features not commonly associated with a pathological condition, and in some cases considered indicative of an overall robust metabolic condition. These include a reduction in cholesteryl esters and intermediates of protein glycation, and an increase in antioxidants and metabolites associated with hypoxia and longevity. This work suggests that certain metabolic features associated with human pathologies are either not intrinsically harmful, or can be counteracted by reciprocal adaptations. We provide a transparent pipeline for reproducing our analysis and a Shiny app for other researchers to explore and visualize our dataset.

## Editor's evaluation

Medley et al., study *A. mexicanus*, an extreme-adapted organism with important connections to human health. The authors test metabolic responses in this natural model of elevated blood glucose and extensive body fat deposits, conditions generally expected to predispose to a higher risk for metabolic syndrome and higher frailty. The work is rigorous and will provide an important reference for future studies aimed at dissecting the mechanistic basis underlying metabolic shifts in this uniquely attractive model. The authors also provide an open and accessible window into their data and analyses by sharing a Shiny app.

## Introduction

Metabolism plays a central role in many cellular processes, and its dysregulation is a hallmark of many disease states, including cancer, obesity, and diabetes. Recent work (*Cirulli et al., 2019*) has shown that certain health effects, particularly cardiovascular disease, can be predicted from metabolic signatures prior to clinical manifestations. This suggests that metabolic dysregulation has causal influence over the disease state of an organism, and conversely disease may be preventable via metabolic intervention (*Rubino et al., 2016*).

An evolutionary system with particularly extreme changes in metabolic regulation is the Mexican tetra, *Astyanax mexicanus*, which has undergone considerable physiological and behavioral changes to colonize a number of subterranean caves in the Sierra de El Abra region of Mexico. Cavefish have evolved a suite of metabolic phenotypes to cope with the cave environment, including lower metabolic rate, increased appetite, fat storage, and starvation resistance (*Aspiras et al., 2015*; *Hüppop, 1986*; *Xiong et al., 2018*). Cavefish are also insulin resistant, hyperglycemic, and exhibit increased caloric intake (*Riddle et al., 2018*), a feature often associated with decreased longevity. A notable genomic feature in Pachón and Tinaja cavefish is a mutation in the insulin receptor (*Riddle et al., 2018*) that, in humans, is linked to Rabson-Mendenhall (RM) syndrome, a form of severe insulin resistance that causes many developmental abnormalities and typically progresses to ketoacidosis (*Longo et al., 1999*). Nevertheless, cavefish do not appear to suffer any of the adverse effects of RM, lack advanced glycation end products (AGEs) (*Riddle et al., 2018*) normally associated with hyperglycemia, and live long, healthy lives without ill-effects of metabolic disease (*Riddle et al., 2018*). *A. mexicanus* may provide natural solutions to overcome the challenges associated with adverse metabolic conditions (*Krishnan and Rohner, 2019*).

From an evolutionary standpoint, survival in the cave environment requires resistance to long periods of nutrient deprivation. This, in turn, leads to storage of excess energy in fat and glycerol, which are themselves potentially harmful to the host organism. We hypothesize that survival in the cave environment thus requires multiple, counterbalancing evolutionary changes and that the combined effect of these changes is to make cavefish resilient to a variety of metabolic conditions, of which starvation and triglyceride / sugar accumulation are discrete examples.

We thus sought to characterize the metabolic signature of resilience by examining the metabolome of Pachón and Tinaja cavefish (two recently derived cave populations) compared to surface-dwelling populations using untargeted mass spectrometry (MS) of primary metabolites and lipids. We characterized the response of energetically important tissues (the liver, muscle, and brain) of each population in short/long-term fasted, and fed conditions. We demonstrate that metabolite profiles in Pachón and Tinaja cavefish are more similar to each other than surface fish in each feeding state / tissue combination, highlighting the role of parallel evolution in shaping the metabolome of cavefish. We identify metabolic signatures and metabolites exhibiting the most pronounced regulatory changes within each tissue, population, and feeding state. We constructed inter-population and inter-feeding state comparisons and fit separate statistical models to each case (*Figure 7—figure supplement 1*). Cavefish exhibit many similarities with metabolic syndrome, but also differ from these conditions in terms of antioxidants, metabolites associated with cellular respiration / the electron transport chain, and unexpectedly reduced levels of cholesteryl esters.

Our results lay the groundwork to explore the mechanistic roles of metabolites and pathways in the nutrient availability-related adaptations of cavefish and suggest that natural evolutionary systems may offer insights into metabolic function by showing how disease states can be altered or counterbalanced under a genetic background more suited to a different set of parameters governing metabolic state.

In order to aid others in reproducing our analysis, we have provided a pipeline to reproduce all major figures and results on this paper that are derived from our metabolomics dataset at https://www.stowers.org/research/publications/libpb-1699 (ftp://odr.stowers.org/LIBPB-1699). We also provide a shiny app at https://cavefin.shinyapps.io/shiny to allow others to explore and visualize our dataset.

Our experimental design aimed to (a) characterize the response of the *A. mexicanus* metabolome to different feeding states in energetically expensive tissues and (b) utilize comparisons across populations and feeding states to identify metabolites conserved in cavefish populations. Food scarcity is one of the cave's harshest evolutionary pressures. Cavefish have specialized feeding strategies and fat metabolism that helps them thrive in the cave environment (*Jeffery, 2020*; *Aspiras et al., 2015*;

**Figure 1.** Metabolic resilience — survivability under a variety of extreme conditions. Certain populations of cavefish have adaptations that cause increased appetite (*Aspiras et al., 2015*) and increased fat accumulation (*Xiong et al., 2018*) (in cases where nutrients are plentiful, such as in lab-raised populations). These same populations also exhibit robust health and longevity (*Riddle et al., 2018*; *Xiong et al., 2018*) and do not suffer ill-effects due to high levels of visceral fat and hyperglycemia, both of which are features of most cave populations. However, visceral fat accumulation in cave populations is highly dependent on nutrient availability and is not displayed in wild-caught specimens (*Krishnan et al., 2020*). Thus, cavefish paradoxically appear to tolerate both extremely low and extremely high levels of triglycerides, glucose, and other energy storage metabolites. We argue that these differences can be reconciled under a hypothesis whereby the cave environment selects not for resistance to nutrient deprivation per se, but rather *resilience* to a variety of nutrient availability states (such as seasonal floods). Survival under such challenging conditions ostensibly favors the ability to tolerate extreme metabolic states, including not only starvation but also high levels of potentially deleterious metabolites such as triglycerides and reactive oxygen species (ROS). We find evidence for elevated antioxidant levels and altered cholesterol / cholesteryl ester homeostasis in cavefish, suggesting that cavefish may use these mechanisms to offset potentially harmful metabolites and tolerate a broad range of metabolic conditions.

*Hüppop, 1986*; *Xiong et al., 2018*). We raised age-matched offspring of Surface (river) fish, and Pachón and Tinaja cavefish morphs originating from two independent cave colonizations. To understand how the cavefish metabolome adapts to ecologically relevant food challenges, we separated Surface, Pachón and Tinaja populations into three different groups at 4 months: 30 day fasted, 4 day fasted, and "Refed" (fed at 3 hr prior to collection after 4 days without food) (*Figures 1 and 2*). We also show RNA-Seq validation (*Figure 6—figure supplement 1*, *Figure 6—figure supplement 2*, *Supplementary file 1*-table s2) of the main themes observed from the metabolomic data.

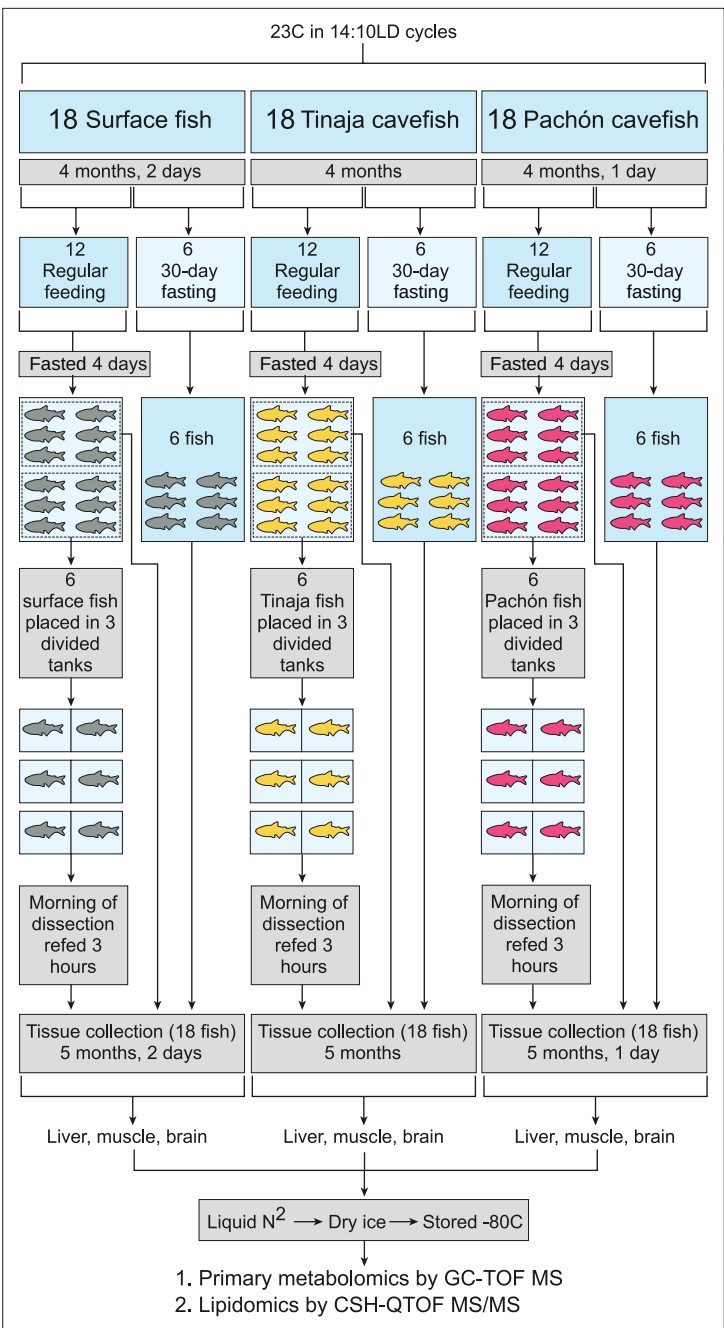

**Figure 2.** Experimental setup for each tissue and metabolite category. Pachón, Tinaja, and surface *A. mexicanus* fry were raised for 4 months and then separated evenly into fasted (30–day) and non–fasted groups. At 4 days prior to collection, non–fasted fish were again divided into two groups (6 fish each) and either fasted for the remaining 4 days (first group) or fasted for 4 days and refed 3 hr prior to collection (second group). Thus, six fish were obtained for each of the following conditions: 30–day fasting, 4–day fasting, and 4–day fasting followed by re–feeding.

## Results

Our untargeted metabolomics study yielded a total of 174 identified metabolites linked to KEGG (*Kanehisa and Goto, 2000*) / HMDB (*Wishart et al., 2018*) IDS and 483 identified lipids linked to LIPIDMAPS IDS. We examined the effect of normalizing identified peak values by the total sum of peaks (mTIC) and by sample weight (Fig supplement 2) and found that mTIC is more robust to variations in sample weight. Hence, we employed mTIC-normalized data for the remainder of the analysis.

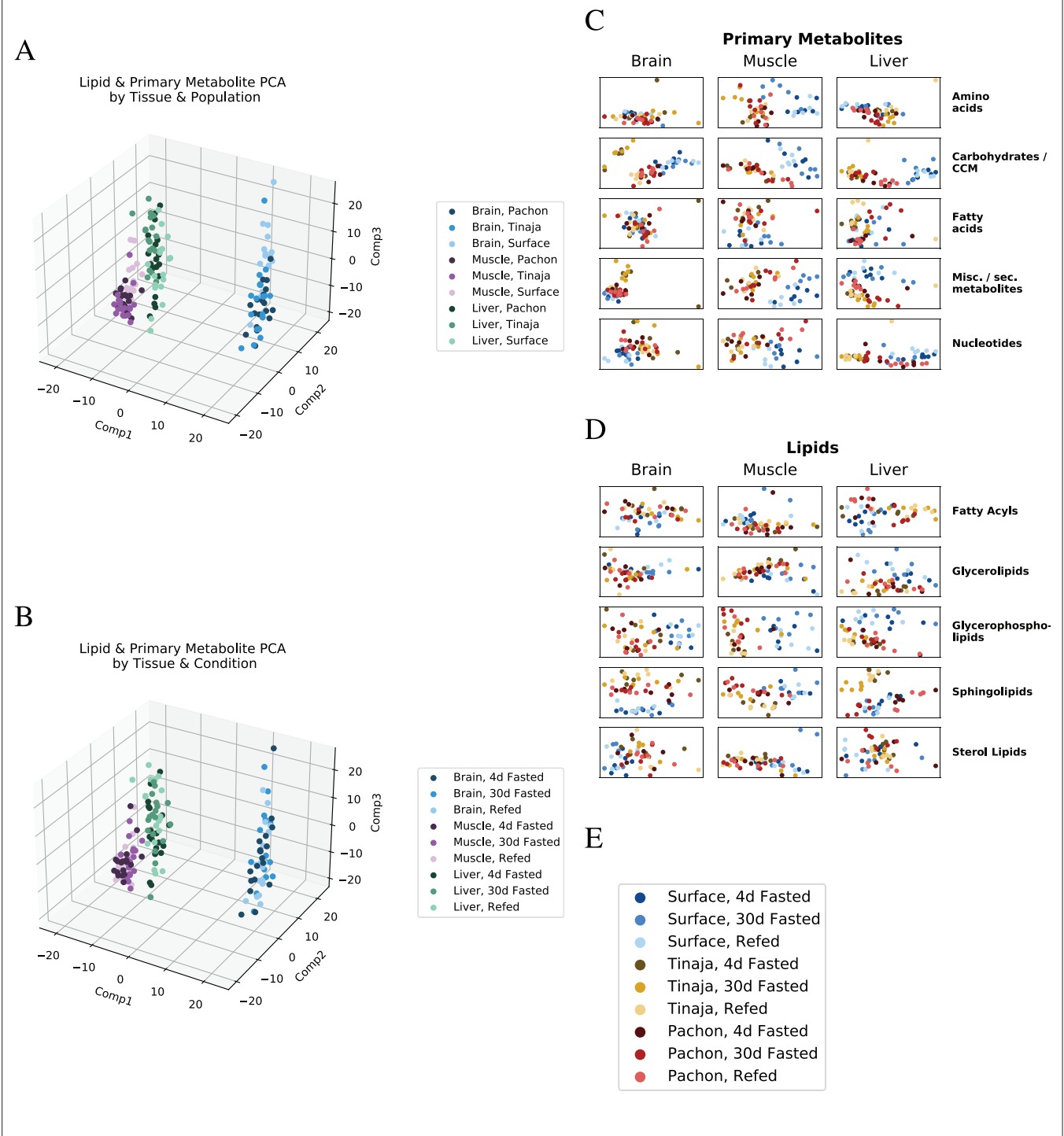

**Figure 3.** Global trends in lipid and primary metabolite data. To visualize overall patterns in the metabolome of different experimental groups, we performed principal component analysis (PCA) first on all lipids and primary metabolites (**A,B**), then on individual categories thereof (**C,D**). Samples tend to cluster primarily by tissue of origin, in line with studies from mammals (***Ma et al., 2015***). Shading is by population (**B**) or feeding state (**C**). Categorical breakdown of primary metabolites (**C**) and lipids (**D**) reveals that cavefish tend to cluster closer to one another than to surface. (**E**) Legend for C and D.

***Figure 3A/B*** shows that metabolites cluster primarily by tissue, in line with previous studies in mammals (***Ma et al., 2015***). ***Figure 3C/D*** shows how clustering patterns depend strongly on the chemical classification of identified metabolites. Some lipid and primary metabolite categories show a clear separation between different populations (e.g. carbohydrates, ***Figure 3C***, and glycerophospholipids,

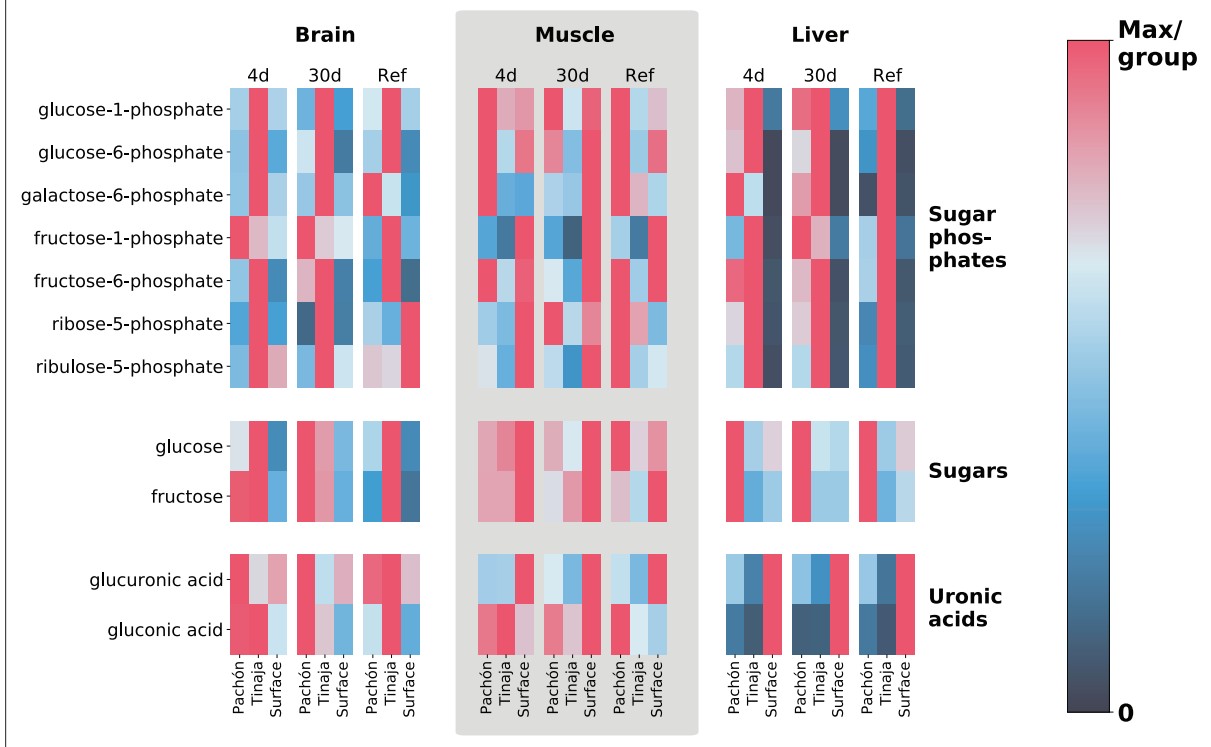

**Figure 4.** Extreme alterations to sugar metabolites in cave vs surface populations. To show differences in cavefish sugar metabolism, we selected 5- and 6-carbon sugars, sugar phosphates, and uronic acids (oxidized forms of sugars that form building blocks of proteoglycans). The row axis shows metabolites within each of these classes, and the column axis shows different populations (short range, labels at bottom), feeding states (medium-range, labels at top) and tissues (long-range, labels at topmost point). Color indicates the mTIC-normalized peak intensity for the average of six biological replicates. Red indicates the maximum value for a given row (i.e. across all populations, tissues, and feeding states), whereas navy blue (the bottom of the color bar) corresponds to a peak intensity of zero, not to the minimum value. Thus, dark cells correspond to very little / zero intensity, as opposed to simply corresponding to the minimum intensity within the row. Given that mTIC intensities are 'semi–quantitative', red values thus correspond to the most abundant group for a given metabolite, and navy blue corresponds to lack of abundance.

*Figure 3D*, across most tissues), whereas other categories have a less pronounced change (amino acids in the brain, *Figure 3C*, and fatty acyls in most tissues). In order to quantify separation of feeding states as a function of population and metabolite category, we used a supervised machine learning method based on orthogonal projection of latent structures (O–PLS, *Figure 7—figure supplement 3*). We then used O–PLS to remove 'orthogonal' variation (*Trygg and Wold, 2002*) from each metabolite category and fit a Bayesian logistic regression model to the de-noised data (Supplemental Methods).

In general, the metabolome of all three populations shows a large degree of similarity within a given tissue (*Figure 7—figure supplement 4*), highlighting the influence of evolutionarily conserved functions of individual tissues.

## Sugar phosphate metabolism

Given the overall similarity at the tissue level for most classes of metabolites (*Figure 7—figure supplement 4*), phenotypic differences are likely to be linked to a relatively small subset of the metabolome. We sought to identify metabolites that could be responsible for the drastic change in phenotype of cave populations. Sugars and sugar phosphates are important energy metabolites and hence candidates for adaptations related to resistance to nutrient deprivation. This class of metabolites displays a dramatic change during short- and long-term fasting, particularly in the liver (*Figure 4*). Transcriptomics data from multiple studies by our group also displays an overall trend toward upregulation of sugar metabolism in cavefish (*Figure 6—figure supplement 1*).

Hepatic glucose production is derived from gluconeogenesis and glycogenolysis, the latter relying on stored glycogen, which is quickly exhausted during fasting (*Han et al., 2016*), indicating that hepatic gluconeogenesis likely plays a role in sustaining survival under long-term nutrient deprivation

in *A. mexicanus*. Surprisingly, surface fish also show stable (albeit generally lower) sugar levels in the liver under different feeding states (*Figure 4*), indicating that sugar production in the liver may be driven by overall demand rather than supply. This may point to a shift from oxidative to sugar-based metabolism as an energy source in energetically expensive tissues. Cavefish possess a larger amount of body fat (*Xiong et al., 2018*; *Aspiras et al., 2015*), and hence have a larger pool of glycerol to serve as a substrate for gluconeogenesis. We find that both cave populations exhibit decreased levels of glycerol in the 30-day fasted state, particularly in Tinaja (*Supplementary file 1*), indicating increased consumption of this intermediate as a substrate for gluconeogenesis may be the source of increased sugar / sugar phosphate abundance in cave populations.

Regardless of the substrates leading to sugar metabolite accumulation, it is clear that a large difference exists between cave and surface populations within this class of metabolites. However, the specifics of this alteration to sugar metabolism appear to be population-specific, with Tinaja showing a large increase in sugar phosphates and Pachón showing an increase in unphosphorylated sugars respectively in short/long-term fasted states in comparison to surface (*Figure 4*).

Other tissues show a mixed response, with muscle displaying increased levels of most sugar phosphates in Pachón but decreased levels of fructose-1-phosphate in both cave populations with respect to surface. The brain displays low levels of sugar / sugar phosphate metabolites overall but possesses increased sugar metabolite abundance in cave populations for certain metabolites and feeding states (*Supplementary file 1*).

While the levels of most simple sugars and sugar phosphates are increased in cavefish with respect to surface, the levels of gluconic acid and glucoronic acid show the opposite pattern (*Figure 4*). Gluconic acid and glucoronic acid belong to uronic acids, a class of sugar acids that are major building blocks of proteoglycans. It has previously been observed that *A. mexicanus* cave morphs lack advanced glycation end products (*Riddle et al., 2018*), which are a defining feature of diabetes and are normally associated with chronic hyperglycemia in humans. Altered metabolism of sugar acids in cavefish may play a role in inhibiting excessive protein glycation and the adverse health effects thereof.

## Ascorbate

A highly unexpected and unexplained feature of our analysis is the abundance of vitamin C, particularly in its oxidized form dehydroascorbic acid (DHAA), across all tissues in cave populations (*Figure 5*). Ascorbic acid (AA), the reduced, active form, is also more prevalent in muscle tissue. DHAA can be recycled back to AA using reducing cofactors such as NADH and NADPH, which can in turn be regenerated from the pentose phosphate pathway and TCA cycle using simple sugars (which cavefish possess in great abundance). For this reason, vitamin C content in food labeling is usually reported as the sum of AA and DHAA (*Wilson, 2002*). Thus, cavefish possess a larger total 'pool' of vitamin C (including interconvertible oxidized and reduced forms, *Figure 5*).

Many cavefish populations exhibit increased appetite and carry an allele of the melanocortin 4 receptor that predisposes them to hyperphagia (*Aspiras et al., 2015*). The increased appetite could cause cavefish to consume more overall food, which could be responsible for the AA/DHAA increase in the refed state. However, this does not pertain to the 30-day fasted state, where AA/DHAA levels are also elevated across all tissues. There is widespread consensus that teleosts, like humans, lack the ability to produce AA endogenously due to the absence of gulonolactone oxidase, which catalyzes the final step in AA biosynthesis (*Ching et al., 2015*). In humans, this enzyme is a pseudogene, whereas in teleosts the gene is absent entirely, thought to be lost in the distant evolutionary past. Thus, the additional AA/DHAA supply likely comes from selective reuptake in the kidney, a process that also occurs in humans to conserve AA/DHAA, or it may be produced by commensal microbiota in cavefish. Trace amounts of AA/DHAA in the feed used in the aquatics facility used to house the fish in this experiment may recirculate throughout the water filtration system and be redistributed to all tanks, including those housing fish in the fasted groups. Nevertheless, it remains that even in the case of circulating trace amounts of AA/DHAA, cavefish appear to exhibit selective retainment of AA/DHAA in larger quantities.

The advantages of AA conservation in adaptation to an environment where prolonged starvation is common are self-evident. AA is involved in collagen formation, and its deficiency leads major loss of integrity of connective tissue. Thus, the ability to retain what little ascorbate is present in underground cave environments would confer an enormous survival advantage to fish.

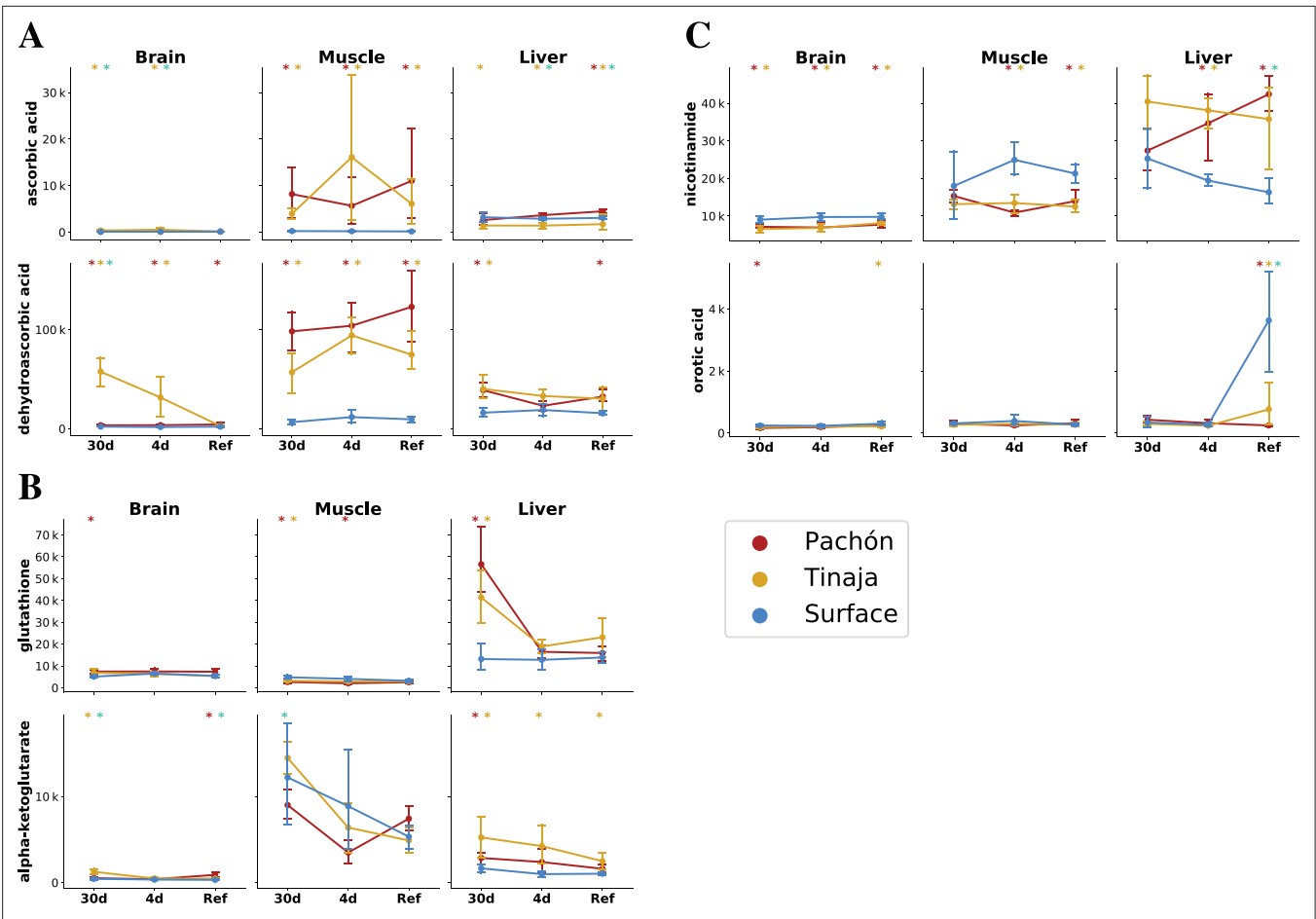

**Figure 5.** The cave and surface fish metabolome highlights differences in antioxidant availability and oxidative metabolism. To analyze changes in redox metabolism in cave populations, we compared the abundance of potent antioxidants and redox-linked metabolites. Metabolites are plotted as mTIC-normalized peak intensities (A,B,C, error bars indicate 2.5/97.5 percentiles). Asterisks indicate significance at the 0.05 level according to an O-PLS / Bayesian logistic regression (Methods) for Pachón vs surface, Tinaja vs surface, and Pachón vs Tinaja. (**A**) Ascorbate (vitamin C) is a potent antioxidant and essential nutrient. Vitamin C exists as the reduced form ascorbic acid (AA) and oxidized form dehydroascorbic acid (DHAA), which can interconverted by cellular processes. (**B**) Glutathione, another antioxidant, is significantly increased in the liver and brain under 30-day fasting. Alpha-ketoglutarate is a tricarboxylic acid cycle (TCA) intermediate that has been linked to longevity in nematodes and mice (***Chin et al., 2014***; ***Asadi Shahmirzadi et al., 2020***). (**C**) Nicotinamide is a precursor to NAD+ synthesis via a salvage pathway, and increased in the liver but decreased in other tissues in cavefish. Orotic acid is a metabolite that causes fatty liver disease in rats when added to a chow diet.

The online version of this article includes the following figure supplement(s) for figure 5:

**Figure supplement 1.** Metabolome analysis heatmaps.

**Figure supplement 2.** Metabolome analysis heatmaps.

Another factor that could influence the AA/DHAA ratio is the effect of insulin resistance and hyperglycemia on the GLUT family of transporters, particularly GLUT4 in adipose/muscle tissue (***Wilson, 2002***; ***Huang and Czech, 2007***). DHAA competes with glucose for transport across the membrane by GLUT4, whereas AA is taken up by Na+ transporters. GLUT4 activity is dependent on membrane translocation and this process is dysregulated in diabetes (***Jaldin-Fincati et al., 2017***). This combination of elevated blood sugar and insulin resistance suggests that GLUT4 could be less active in cavefish and cause DHAA to accumulate in the extracellular space. AA/DHAA have also been reported to influence C-reactive protein levels (***Ford et al., 2003***) and protein glycation (***Franke et al., 2013***), prompting further investigation into *A. mexicanus* as a model of diabetes-related resilience. Finally, Pachón cavefish possess a reduction in neutrophils, one cell type which are normally involved in the uptake of AA and reduction to DHAA, compared to surface (***Peuß et al., 2020***).

## Adaptation to hypoxic conditions

Energy metabolism in most organisms can be viewed as a balance between oxidative processes (cellular respiration via oxidative phosphorylation and the electron transport chain) and sugar metabolism, and the relative contributions of these two processes can have important physiological consequences, as in the well-known Warburg effect in cancer. *A. mexicanus* cave morphs have considerably upregulated sugar metabolism (*Figure 4*), and also display decreased levels of several products of oxidative metabolism. One important metabolite in this category that displays differences in cave populations is $\alpha$-ketoglutarate ($\alpha$-KG), which has increased abundance in the liver in all feeding states and in the brain in certain feeding states in both cave populations. $\alpha$-KG supplementation has been linked to lifespan extension in *C. elegans* (*Chin et al., 2014*) and mice (*Asadi Shahmirzadi et al., 2020*). Furthermore, uronic acids, the oxidative products of simple sugars, are significantly reduced in the liver of both cave populations (*Figure 4*, *Supplementary file 1*), suggesting that cave fish are characterized by decreased reliance on oxidative metabolism and increased reliance on sugar metabolism.

## Obesity and inflammation-related metabolites

Chronic inflammation of adipose tissue is a common feature of obesity and can often lead to insulin resistance and eventually type 2 diabetes (*Paschoal et al., 2020*). Cave populations of *A. mexicanus* have been previously reported to exhibit pronounced insulin resistance (*Riddle et al., 2018*), but do not accumulate advanced glycation end products and do not appear to have diminished longevity.

In order to compare the metabolome of *A. mexicanus* cave populations to the known metabolic signatures of obesity (*Cirulli et al., 2019*), we calculated changes in lipid categories (the coarsest abstraction used in LipidMaps), classes (a more detailed partitioning scheme used in LipidMaps), and, within free fatty acids specifically, the degree of saturation. The metabolome displays a remarkable overlap with the proinflammatory signature associated with obesity that, in humans, leads to insulin resistance. This signature consists of *Aspiras et al., 2015* the elevation of saturated fatty acids (SFAs) in muscle in most feeding states, which have a direct and pronounced proinflammatory effect in mammals through the recruitment of macrophages (*Glass and Olefsky, 2012*), although the importance of fatty acid release in insulin resistance is disputed (*Morigny et al., 2019*; *Karpe et al., 2011*; *Xiong et al., 2018*) abundance of ceramides in muscle in all feeding states, which are known direct mediators of insulin signaling (*Glass and Olefsky, 2012*). Indeed, the only feeding state for which skeletal muscle did not display increased SFA abundance was 30-day fasting, which could simply indicate the exhaustion of free SFA pools. Additionally, palmitate, a precursor of ceramide biosynthesis (*Glass and Olefsky, 2012*), is elevated in muscle in all feeding states. Sphingoid bases are significantly more abundant in muscle in all feeding states, suggesting generally upregulated sphingolipid biosynthesis in cave populations.

In contrast to proinflammatory metabolites, omega-3 fatty acids ($\omega$-3 FAs) such as DHA and EPA have protective effects against inflammation (*Paschoal et al., 2020*; *Glass and Olefsky, 2012*; *Oh et al., 2010*). These molecules bind to the GRP120 receptor on macrophages and adipocytes, and the activated receptor then modulates the activity of PPARγ and ERK (*Paschoal et al., 2020*; *Oh et al., 2010*). $\omega$-3 FAs are less abundant in the liver under 4-day fasting and are generally not upregulated in most feeding states and tissues. Thus, $\omega$-3 FAs do not appear to offset for the proinflammatory signature of cavefish SFA and ceramide signatures, suggesting that cavefish possess an alternate compensatory mechanism to prevent chronic tissue inflammation. Overall, cavefish appear to exhibit many metabolic similarities with obesity and health conditions associated with it.

However, this is not a universal trend. Cirulli et al. report a strong association between urate levels and BMI, likely due to insulin resistance interfering with uric acid secretion in the kidney *Cirulli et al., 2019*. In contrast, cavefish appear to have significantly reduced levels of uric acid in muscle, and in other tissues levels are comparable with surface except for a small but significant increase in Pachón liver during fasted states. Mannose, which is associated with obesity and insulin resistance (*Cirulli et al., 2019*), was abundant in the Pachón liver in all feeding states, but was reduced in Tinaja compared to surface fish.

Finally, cholesteryl esters and cholesterol in some feeding states (Table *Supplementary file 1*), were less abundant in cave populations. Using a previously published gene expression dataset (*Krishnan et al., 2022*), we investigated factors that might influence levels of cholesteryl esters. Cholesteryl

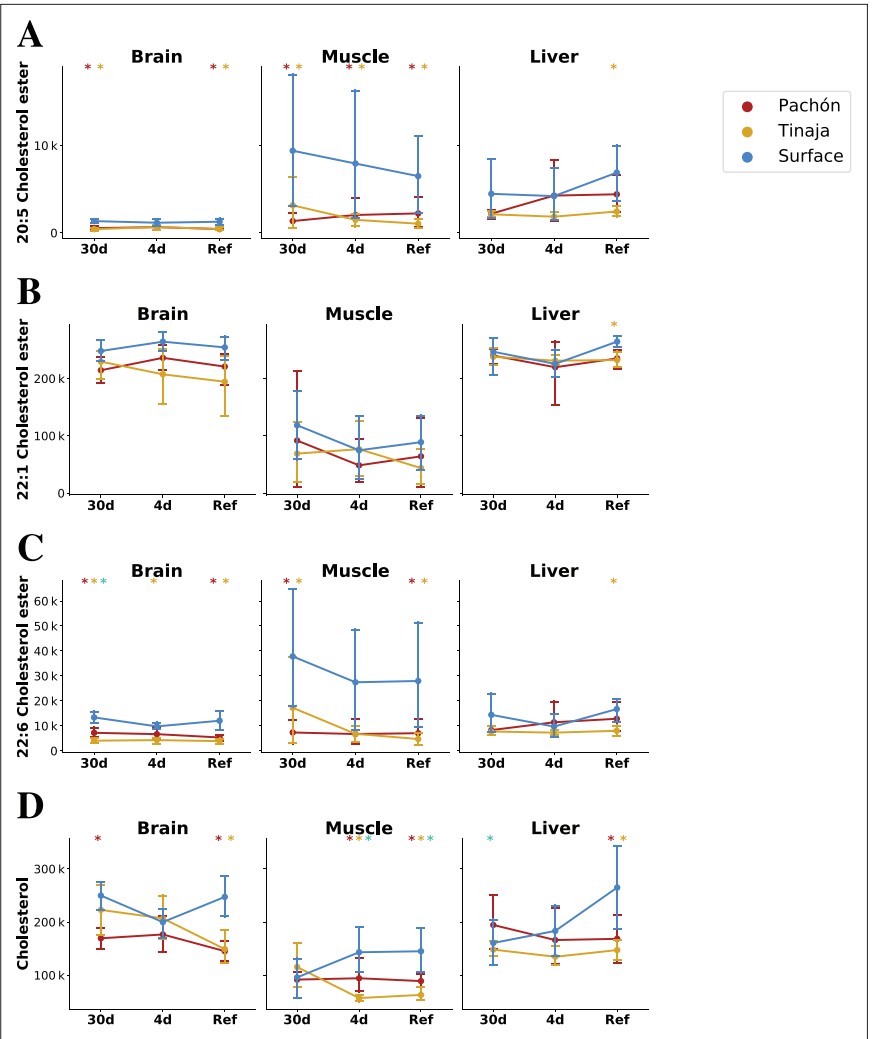

**Figure 6.** Parallel evolution in cavefish exhibits profound alteration of cholesterol / cholesteryl ester metabolism. Cavefish possess a significant reduction in certain long-chain fatty acid cholesteryl esters (**A–C**) and cholesterol itself (**D**), particular in peripheral tissues (muscle). Values on the y–axis are mTIC-normalized peak intensities for each lipid species. Asterisks indicate significance at the 0.05 level according to an O-PLS / Bayesian logistic regression (Methods) for Pachón vs surface, Tinaja vs surface, and Pachón vs Tinaja.

The online version of this article includes the following figure supplement(s) for figure 6:

**Figure supplement 1.** Metabolic themes of *A. mexicanus* cave adaptation revealed by three independent RNA-Seq experiments.

**Figure supplement 2.** Profile of sugar / energy metabolism-related gene expression.

ester transfer protein (CETP), which transports cholesterol in / out of lipoproteins, is downregulated in both Pachón and Tinaja compared to surface fish (*Figure 7—figure supplement 5*), suggesting a potential causal relationship between lower cholesteryl ester levels in cave populations and this important carrier protein. A human variant of CETP associated with decreased serum levels of the protein and larger low-density lipoprotein (LDL) and high-density lipoprotein (HDL) particle sizes has been linked to exceptional longevity (*Barzilai et al., 2003*). The LDL / HDL cholesterol ratio, mediated in part by CETP (*Christison et al., 1995*), is a major contributor to risk of atherosclerosis and coronary heart disease (*Brousseau et al., 2004*). The reliance on triglycerides as an energy source in cavefish may increase the risk of arterial disease by providing an abundance of free fatty acids and other lipids. Cholesteryl esters, in particular, are formed from esterification of a fatty acid and cholesterol, are a major constituent of foam cells in atherosclerotic lesions (*Ghosh et al., 2010*; *Yu et al., 2013*), and, for certain lipid species, show a large difference in abundance between cavefish and surface (*Figure 6*).

Inhibition of CETP has been shown to reduce cardiovascular risk, ostensibly by altering the partitioning of cholesteryl esters between LDL and HDL (**Barter and Rye, 2012**). Indeed, lipid homeostasis in zebrafish exhibits strong similarities with human (**Fang et al., 2014**), and zebrafish express CETP whereas other model organisms such as mice do not. This raises the question of whether CETP and its cholesteryl ester substrates may be a locus under selection in cavefish to offset the risks associated with increased visceral fat.

Large differences in CETP expression suggest that lipid homeostasis may be regulated differently between cave and surface populations. Lipoproteins, in the form of HDL, IDL, LDL, VLDL, and chylomicrons contribute to and regulate lipid homeostasis.

In summary, lipid metabolism in *A. mexicanus* represents a hub of evolutionary activity which clearly separates surface and cave populations. Cavefish must balance increased demands on energy storage with counter-adaptations to protect against pro-inflammatory and atherogenic metabolites. We observe elevated levels of most energy metabolites, with the notable exception of cholesterol and cholesteryl esters, and cavefish have a larger (V)LDL/HDL ratio as compared to surface fish. LDL and VLDL have higher triglyceride content as compared to HDL (**Cox and García-Palmieri, 2011**) and VLDL is a substrate for lipoprotein lipase (**Freeman and Walford, 2016**), which liberates free fatty acids from triglycerides. LDL/VLDL may thus be important for energy metabolism in cavefish, and hence selective pressure may contribute to the higher (V)LDL/HDL ratio in this population.

## Resistance to nutrient deprivation

In order to determine the basis of cavefish adaptation to low-nutrient environments, we sought a statistical test that would be sensitive to metabolites that change significantly between refed and short/long-term fasted states and insensitive to metabolites that remain relatively stable across feeding states. We further hypothesized that certain metabolites may have an important role in cave adaptation. Pachón and Tinaja represent independently derived populations, and we reasoned that a test for parallel adaptation should be selective for metabolites that show the same differential feeding state response pattern across cave populations (e.g. differentially increased in both Pachón and Tinaja fasted states relative to surface). In order to construct this test, we fitted a Bayesian GLM (**Gelman et al., 2008**) to a linear combination $(P + T)/2 - S$ of O-PLS-filtered z-score values (see Materials and methods), where $P$ stands for Pachón, $T$ stands for Tinaja, and $S$ stands for surface. We used this test to identify metabolites that might have a role in the fasting response of cavefish, that is metabolites that are differentially abundant in cave populations in the fasted versus refed state (Pachón and Tinaja are assigned equal weight), and generally show the opposite pattern in surface. **Figure 7** shows the results of this test for 30-day fasting vs. refeeding (A, which corresponds to the most extreme experimental groups), and the two other possible comparisons between feeding states (B/C).

Sugar metabolites do not appear to exhibit a strong differential feeding state response. However, long-chain fatty acids such as palmitate and stearate (**Figure 7C**) do show differential abundance between long- and short-term fasting, suggesting that cavefish may rely on increased usage of fat stores in long-term fasting. Furthermore, analysis of the 30-day fasting response in cavefish liver highlights orotic acid (OA, **Figure 7A**), an intermediate in pyrimidine synthesis that has been implicated in fatty liver condition (**Standerfer and Handler, 1955**). OA is suppressed in all feeding states in cave populations, but exhibits a sharp spike in refed surface fish (**Figure 5**).

Starvation has detrimental effects on an organism in many ways. One detrimental effect is the depletion of antioxidant substances and the resulting oxidative stress through increasing levels of reactive oxygen species (ROS) (**Furne and Sanz, 2017**). Studies that focus on the impact of food deprivation on oxidative stress in fish show that prolonged starvation decreases the capacity of fish to ameliorate oxidative stress (**Furne and Sanz, 2017**). Glutathione is a major antioxidant that detoxifies ROS and thereby prevents cellular damage from oxidative stress (**Pompella et al., 2003**). Cavefish face prolonged periods of nutrient deprivation in their natural environment (**Aspiras et al., 2015**). Adaptation to the cave environment may have led to changes in glutathione metabolism in cavefish to protect against oxidative stress under prolonged fasting. Indeed, in an earlier study we were able to demonstrate that cavefish show an increased expression of genes that are involved in the metabolism of glutathione, which is indicative of an increased stress level compared to surface fish in their natural habitat (**Krishnan et al., 2020**).Here, we can confirm that these trends in gene expression are accompanied by elevation of reduced glutathione in the liver and brain (**Figure 5**, **Supplementary**

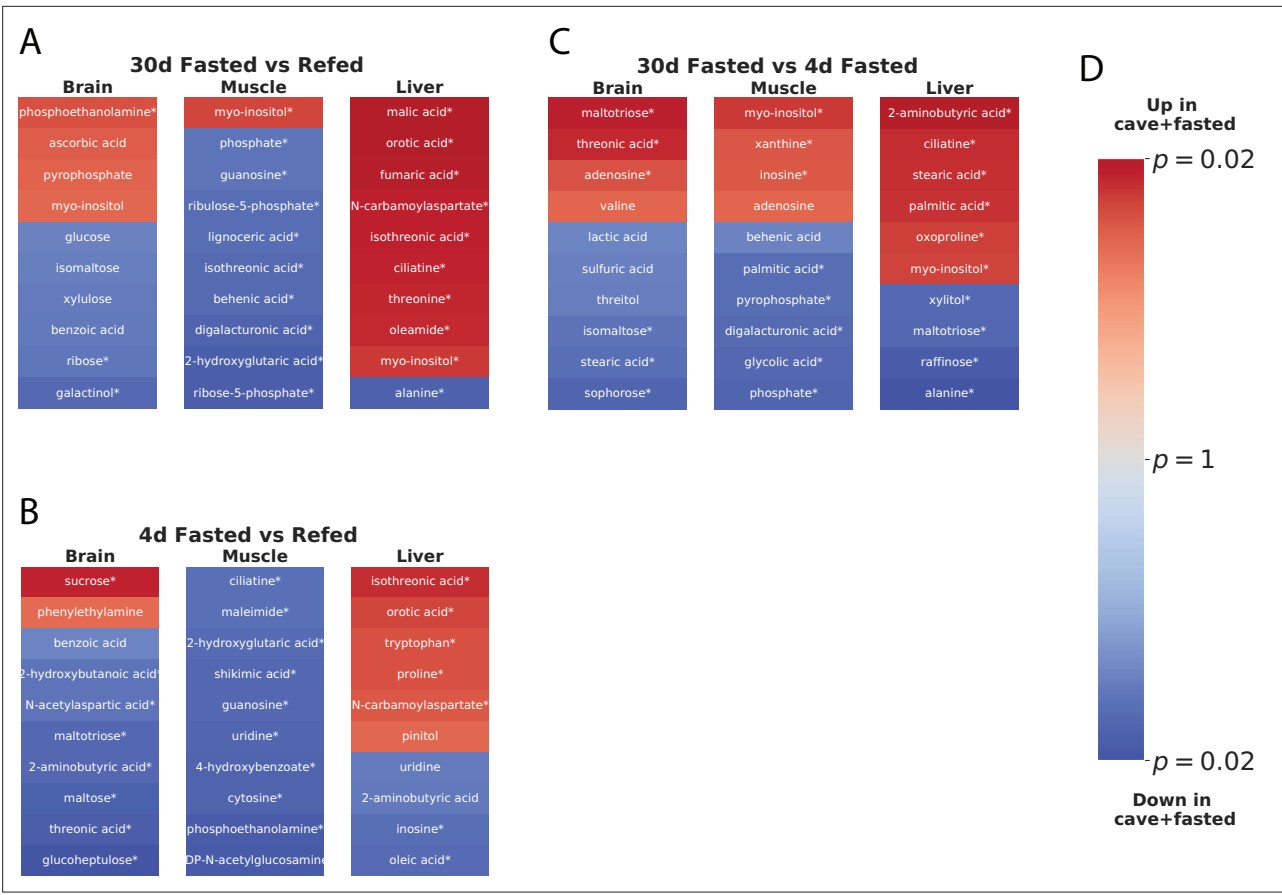

**Figure 7.** Parallel adaptive metabolic signature in response to food deprivation in cave populations. To identify metabolites linked to adaptations promoting survival in a nutrient-limited environment shared between cave populations, we fit an O-PLS / GLM statistical model to the response to fasting, that is the difference between 30-day fasting and refeeding for Pachón, Tinaja, and surface populations. Specifically, we fitted a Bayesian GLM (see Methods) to the linear combination $(P + T)/2 - S$, with $P$, $T$, and $S$ referring to the normalized z-scores for each Pachón, Tinaja, and surface sample respectively. Coloring in the figure indicates metabolites increased (red) or decreased (blue) in both cave populations in 30-day fasting (i.e. 'up' refers to metabolites that are increased in Pachón and Tinaja in the 30-day fasted state with respect to surface), and color intensity corresponds to $log_{10}$ p-value, with lighter colors indicating less significant p-values and darker colors indicating more significance. The most significant 20 differentially abundant metabolites (regardless of direction) in each tissue for both cave populations with respect to surface are displayed. An asterisk (*) indicates significance at the 0.05 level.

The online version of this article includes the following figure supplement(s) for figure 7:

**Figure supplement 1.** Schematic depiction of comparisons used for fitting GLM parameters.

**Figure supplement 2.** Effect of normalization scheme on peak intensity distribution.

**Figure supplement 3.** Classifier performance shows which categories of primary metabolites are most salient in the starvation response.

**Figure supplement 4.** Relative composition for primary metabolites and lipid classes shows metabolome profile for different tissues, populations, and feeding states.

**Figure supplement 5.** CETP, but not HMG-CoA reductase homologs, show identical or reduced expression in surface fish.

file 1). We did not observe a significant increase of glutathione in the surface fish in the fasted states (*Figure 5*, *Supplementary file 1*).

Guided by these observations, we further attempted to characterize the role of altered antioxidant and cholesterol metabolism in cavefish. In particular, we hypothesized that cavefish resilience to widely varying nutrient levels is driven by a robust antioxidant system that prevents the accumulation of ROS under conditions of stress (e.g. induced by fasting).

To test this hypothesis, we examined ROS state in the liver under a subset of our original fasting experiment using similarly aged fish. ROS accumulation has been shown to cause lethal levels of cell damage in flies, but such damage can be prevented via oral administration of antioxidants or

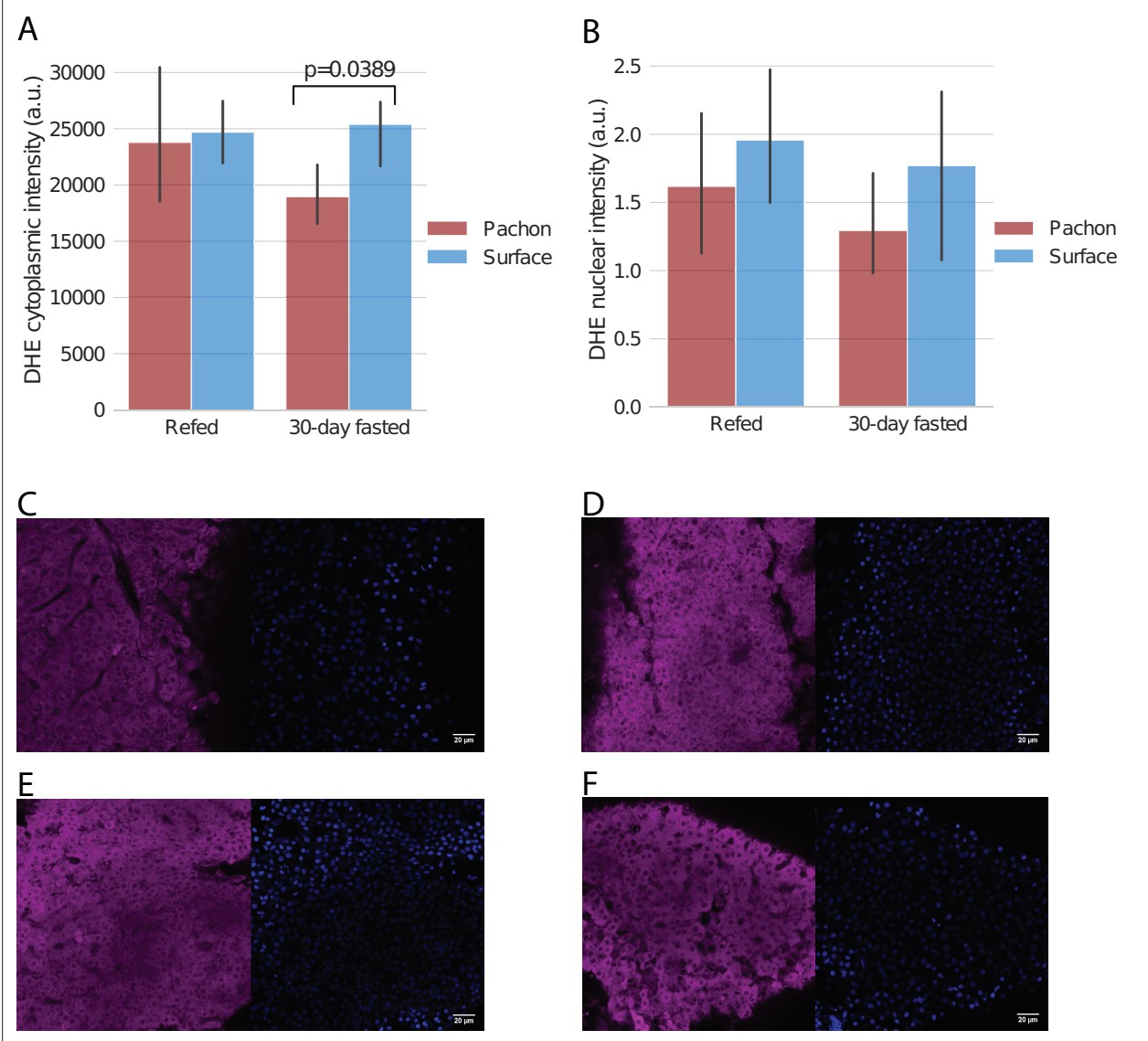

**Figure 8.** ROS levels in refed / fasted liver via DHE staining. Pachón cavefish and surface fish were fasted for 30 days and livers were removed and stained with dihydroethidium (DHE), a blue-fluorescent dye that fluoresces red when oxidized and serves as an indicator of the presence of superoxide radicals. While we did not detect a significant difference in baseline staining in fish refed after 4 days of fasting, cytoplasmic ROS in Pachón was significantly (Kruskal-Wallis test) decreased in Pachón with respect to surface as measured by DHE staining (**A**). Nuclear differences were not detected among any groups (**B**). (**C–F**): Arbitrarily selected images of DHE (left) and DAPI (right) staining for Pachón with 4-day fasting followed by refeeding (**C**), surface with 4-day fasting followed by refeeding (**D**), Pachón with 30-day fasting (**E**), and surface with 30-day fasting (**F**). All images are available at https://www.stowers.org/research/publications/libpb-1699.

over-expression of antioxidant-producing enzymes (*Vaccaro et al., 2020*). This caused us to ask whether naturally elevated antioxidant levels in cavefish might protect against starvation-induced ROS accumulation. We reasoned that the most extreme changes are observed between 30-fasted and 4-day fasted +refed groups, and that given the similarity in Pachón and Tinaja antioxidant profiles, one cave population would be sufficient for validation. We thus repeated the fasting experiment using only 30-day fasted and 4-day fasted +refed groups, and using only Pachón and surface populations.

Using dihydroethidium (DHE) staining, we examined different populations and feeding states for ROS level. While liver sections of fasted / refed fish did not show differences in cytoplasmic / nuclear localization of DHE (*Figure 8B–E*), a quantitative analysis showed that, under 30-day fasting, juvenile Pachón cavefish exhibit lower levels of ROS in cytoplasm as compared to surface under the same conditions (*Figure 8A*), supporting the hypothesis that elevated antioxidants in cavefish may have a

protective effect by neutralizing active ROS and may have arisen as an adaptation allowing cavefish to tolerate ROS produced by long periods of nutrient deprivation. We did not observe significant differences in DHE nuclear intensity for either the cytoplasm or the nucleus, suggesting that 30 days of fasting may not be sufficient to produce salient differences in the nucleus for juvenile fish. The known resilience properties of antioxidants may also help explain the metabolic resilience of cavefish by providing a buffer against different forms of stress.

Cavefish thus exhibit decreased ROS abundance in the liver in 30-day fasted conditions, which correlates with the trend we observed in glutathione. Analysis of three separate transcriptomic experiments conducted by our group also points to important changes in key genes, such as glutathione S-transferase, being consistently upregulated in cavefish (*Figure 6—figure supplement 1*, *Supplementary file 1*-Table S2). Combined with the findings presented here, this suggests that response oxidative stress is indeed an important factor in cavefish physiology.

Antioxidant levels in cavefish may have evolved as a strategy to allow cavefish to tolerate variation in nutrient availability, as a means of controlling inflammation caused by deleterious energy metabolites, as a way of inhibiting atherogenesis, or several of these effects.

## Discussion

*A. mexicanus* has been advanced as a model of resilience under ostensibly pathological conditions including hyperglycemia, diabetes (*Krishnan and Rohner, 2019*), and insulin resistance (*Riddle et al., 2018*). Here, we have provided a large, untargeted study of the metabolome of *A. mexicanus* surface fish and two cave populations in order to investigate the molecular underpinnings of these adaptations.

We were particularly interested in the role of metabolism in cave adaptation of the two *A. mexicanus* cavefish populations in this study: Pachón and Tinaja. This suggests that parallel adaptation to cave environments requires satisfying certain common metabolic needs that are an inherent part of the niche. The obvious candidate for this evolutionary conflux is adaptation to a low-nutrient environment. However, metabolic strategies for survival in such environments are not currently well-understood. We found that drastic alterations in energy metabolism, together with shifts in mediators of redox metabolism and ascorbate, an essential vitamin which is lacking in the cave environment, constitute a major feature of cave adaptation in these populations.

Cavefish appear to have substantially altered sugar metabolism, and exhibit higher levels of sugars and sugar phosphates. However, the opposite trend occurs for uronic acids, which are the oxidized forms of simple reducing sugars and can be formed enzymatically or non-enzymatically. This incongruency can be resolved by noting the overall trend to decreased reliance on oxidative metabolism (and enzymes that catalyze oxidative processes) and increased reliance on sugar metabolism. This trend stems from sugars and sugar phosphates, antioxidants such as ascorbate and glutathione, and $\alpha$-KG (which has been shown to inhibit the electron transport chain in *C. elegans*). Due to drastic fluctuation in oxygen level in the subterranean niche, cavefish may rely on a shift from oxidative to predominantly sugar-derived energy metabolism, as compared to their surface-dwelling cousins. Reduction in uronic acids, which are derived from sugars using oxidative processes, can thus be seen as part of this trend. However, the specific reduction of uronic acids in particular may have an additional survival benefit for cavefish by inhibiting protein glycation and thus preventing accumulation of advanced glycation end products. Further investigation is required to fully understand the evolutionary and physiological implications of these metabolic changes.

Further work is required to establish the extent of hypoxic conditions in *A. mexicanus* evolution. However, we also find that certain redox-related metabolites, including $\alpha$-KG, glutathione, and ascorbate, all exhibit distinctive abundance patterns in cavefish. These patterns may be in response to hypoxia, poor nutrient conditions, differences in metabolic rate, or some other aspect of the cave niche.

Our data indicate that upregulation of glucose and long-chain fatty acid production is a common feature shared by Pachón and Tinaja cave populations, suggesting that certain cave habitats do require considerable changes in energy metabolism. Pachón and Tinaja likely have a greater reliance on fat stores for locomotion, as evidenced by increased SFA content in muscle in fasting and refeeding. The decrease of $\omega$-3 FAs during fasting (*Supplementary file 1*) coupled with the increase of palmitate (*Supplementary file 1*) in long- vs short-term fasting suggests that cavefish metabolism may be

preferentially biased toward storing caloric intake as energy-rich saturated fatty acids. Whether cavefish possess adaptations to counteract the deleterious effects of high body fat, such as suppression of orotic acid (a metabolite implicated in steatosis in the liver), requires further investigation.

In summary, *A. mexicanus* troglomorphic populations share many metabolic similarities with known pathological conditions such as obesity, but also display important differences which may help to explain their resistance to negative effects of these pathologies. We also found important differences in ascorbate, which is known to serve diverse physiological roles, nicotinamide, which is a precursor to $NAD^+$ synthesis and hence is related to oxidative metabolism, $\alpha$-ketoglutarate, which has been implicated in longevity in *C. elegans* (*Chin et al., 2014*). We have shown via ROS staining that Pachòn cavefish do possess lower levels of superoxide radicals in the liver after 30 days of fasting. This confirms that increased antioxide levels in cavefish do indeed correlate with physiological state, and suggests that selection in cave environments favors resistance to oxidative stress. Whether selective pressure is driven chiefly by starvation resistance, or whether it is a mechanism to offset oxidized LDL and other harmful metabolites is an open question.

*A. mexicanus* cave populations clearly exhibit altered lipoprotein composition and expression of CETP, an important component of lipid homeostasis. Together with increased abundance of antioxidants, this may contribute to the ability of cavefish to withstand high triglyceride levels.

Finally, in order to make our data and methods maximally available to other researchers, we have implemented a transparent pipeline that can be used to regenerate all main figures and tables presented here. We have also used the structured data library ObjTables (*Karr et al., 2020*) to provide machine-readable, semantically accurate representations of the results presented here.

This study provides a large, untargeted metabolomics and lipidomics study of *A. mexicanus* surface and cave morphs. However, there are limitations to our approach. Our analysis was based on juvenile, pre-sexually mature fish. We chose this developmental time point under the hypothesis that evolutionary changes in metabolism would tend to act early to respond to the selective pressure of the cave environment and positive juvenile fish for robust starvation resistance. However, an age of 5 months does not necessarily coincide with seasonally-correlated food shortages, and hence manifestations of starvation adaptations may not occur until a later developmental stage. We also did not examine a fasting duration of greater than 1 month, despite the potential for longer periods of nutrient deprivation in cave environments.

## Conclusion

Our goals for this study were (*Aspiras et al., 2015*) to provide a comprehensive untargeted study of primary metabolites and lipids in *A. mexicanus*, an extreme-adapted organism with important connections to metabolic disease, (*Xiong et al., 2018*) examine the molecular basis for low-nutrient adaptation in cave-dwelling subpopulations, and (*Riddle et al., 2018*) identify metabolic changes that might explain *A. mexicanus* longevity in the face of a phenotype with properties linked to metabolic syndrome.

Our findings show that the adaptation to a low nutrient environment in *A. mexicanus* is linked to extreme changes in sugar and fat metabolism, and that increased reliance on these energy sources in the liver provides for the needs of the organism during nutrient availability fluctuations.

All in all, our results highlight the role of *A. mexicanus* as an evolutionary example of extreme metabolism and suggest important roles for certain metabolites in fish and other species.

## Materials and methods
### Experimental model and subject details

Surface morphs of *Astyanax mexicanus* were reared from offspring of Mexican surface fish collected in the Río Choy. Pachón and Tinaja morphs were reared from fish originating from the Pachón and Tinaja caves. A total of 18 fish from each population were used in experiments. Sex was not determined due to difficulties in determining sex in juvenile *A. mexicanus* fish. This study was approved by the Institutional Animal Care and Use Committee (IACUC) of the Stowers Institute for Medical Research under

protocol 2019–084. Animals were euthanized according to an IACUC-approved euthanasia protocols based on American Veterinary Medical Association (AVMA) guidelines using Tricaine mesylate. The method currently in use has been updated to reflect 2020 AVMA guidelines and uses 30 min of opercular movement cessation unless a secondary method is employed.

## Method details

### Fish husbandry

All *Astyanax* are housed in glass fish tanks on recirculating aquaculture racks (Pentair, Apopka, FL) with a 14:10 LD photoperiod. Each rack system is equipped with mechanical, chemical, and biological filtration and UV disinfection. Water quality parameters are monitored and maintained daily as described in previous studies (*Xiong et al., 2018*; *Peuß et al., 2020*). Fish were fed once per day with mysis shrimp and twice per day with Gemma diet. Gemma feed is Protein 59%; Lipids 14%; Fiber 0.2%; Ash 14%; Phosphorus 1.3%; Calcium 1.5%; Sodium 0.7%; Vitamin A 23,000 IU/kg; Vitamin D3 2,800 IU/kg; Vitamin C 1000 mg/kg; Vitamin E 400 mg/kg. Health examinations of all fish were conducted by aquatics staff twice daily. *Astyanax* colonies are screened biannually for ectoparasites and endoparasites and no pathogens were present at the time of this study. Fish treatment and care was approved by the Institutional Animal Care and Use Committee (IACUC) of the Stowers Institute for Medical Research. NR's institutional authorization for use of *Astyanax mexicanus* in research is 2019–084.

### Feeding regimen and tissue collection

Age-matched offspring of Surface, Pachón, and Tinaja populations were reared in similar densities at 23 °C in 14:10LD cycles as described previously. Fish were the result of a group mating event within populations. Fish were housed only with members of their population for their entire lives. At 4 months (Tinaja), 4 months and 1 day (Pachón), 4 months and 2 days (Surface), fish of each population were separated into two tanks. 12 fish were separated and starved for 30 days until tissue collection. The 12 fish were maintained on regular feeding schedules until 4 days prior to tissue collection when food was withheld from each population's regular feeding tank. The mass (g) and length (cm) of each fish was recorded at separation. All efforts were made to equalize mass and length distributions in each cohort. On the evening before tissue collection, 6 fish from the 4-day starved tank were separated and placed into three, 3L-tanks. Tanks were divided down the middle such that all 6 fish (2 in each tank) were housed individually. Singly housed fish were refed for 3 hr with 10 mg of Gemma 500 on the morning of the dissection day for each population. Dissection occurred at 5 months (October 5th, 2019; Tinaja), 5 months and 1 day (October 6th, 2019; Pachón), 5 months and 2 days (October 7th, 2019; Surface). Fish were re-fed in intervals between 8:30am and 12:00pm. At each 3 hr time point, a re-fed fish, a 4-day starved fish, and a 30-day starved fish was euthanized in MS-222. To reduce variability between populations dissected on subsequent days, all dissections took place between 11:30-3pm and were handled identically. Prior to dissection, the final mass and length were recorded for each fish. The liver, muscle, and brain were dissected and placed in 1.5 mL plastic tubes. Tissues were flash frozen on liquid nitrogen, transferred to dry ice and stored at –80 °C. Samples were shipped to West Coast Metabolomics Center on dry ice overnight for analysis.

### Sample preparation

Samples were prepared using the Matyash protocol (*Matyash et al., 2008*). This procedure allows efficient extraction of lipids in a non-polar methyl tert-butyl ether (MTBE) layer, and extraction of primary metabolites in the polar water/methanol layer (*Fiehn, 2016*; *Cajka and Fiehn, 2014*). From each sample, 4.1 mg of frozen liver or brain tissue (+/-0.3 mg) or 10.1 mg of muscle tissues (+/-0.3 mg) was weighed and placed into 1.5 mL Eppendorf tubes. Samples were ground prior to extraction using beads with a Spex Sample Prep GenoGrinder with stainless steel 2–3 mm beads for 30 s. 975 µL of ice cold, 3:10 (v/v) MeOH/MTBE +QC mix/CE (22:1) extraction solvent was added to each homogenized sample. Samples were vortexed for 10 s and shaken for 5 min at 4 °C. A total of 188 µL room temperature LC/MS water was added and samples vortexed for 20 s, then centrifuged for 2 min at 14,000 rcf. The upper organic phase was transferred to two separate tubes (350 µL each) for lipidomics (CSH) analysis. The bottom aqueous phase was transferred to two additional tubes (110 µL each) for primary metabolism (GC-TOF) analysis. One tube from each phase was reserved as a backup, the other tube

was dried down completely using centrivap. Both were kept at –20 °C until ready for analysis. As an additional step prior to GC-TOF analysis, samples were resuspended in 500 µL of degassed, -20°C mixture of acetonitrile (ACN): isopropanol (IPA): water (H2O) (3:3:2, v/v/v). Samples were vortexed for 10 s and then centrifuged at 14,000 rcf for 2 min. A total of 450 µL of supernatant was transferred to a new tube and concentrated to complete dryness using a Labconco Centruvap cold concentrator.

## Primary metabolite data acquisition

Metabolite abundances were quantified by gas-chromatography, time-of-flight mass spectrometry (GC-TOF/MS) using previously established methods (*Fiehn et al., 2008*). Briefly, an Agilent 6,890 gas chromatograph (Santa Clara, CA) equipped with a Gerstel automatic linear exchange systems (ALEX) which included a multipurpose sample dual rail and a Gerstel cold injection system (CIS) was used with a Leco Pagasus IV time-of-flight mass spectrometer running Leco ChromaTOF software. The injection temperature was ramped from 50 °C to a final temperature of 275 °C at a rate of 12 °C/s and held for 3 min. Injection volume was 0.5 µl with 10 µl/s injection speed on a splitless injector with a purge time of 25 s. The liner (Gerstel # 011711-010-00) was changed automatically every 10 samples to reduce sample carryover. The injection syringe was washed three times with 10 µL ethyl acetate before and after each injection. For gas chromatography, a Rtx-5Sil MS column (30 m long, 0.25 mm i.d.) with 0.25 µm 95% dimethyl 5% diphenyl polysiloxane film was used (Restek, Bellefonte PA). The GC column was equipped with an additional 10 m integrated guard column. 99.9999% pure Helium with a built-in purifier was set at a flow rate of 1 mL/min. The oven temperature was held constant at 50 °C for 1 min, ramped at 20 °C/minute to 330 °C, and then held constant for 5 min. The transfer line temperature between gas chromatograph and mass spectrometer was set to 280 °C. The mass spectra were acquired at a rate of 17 spectra/second, with a scan mass range of 80–500 Da at an ionization energy of –70 eV, 1800 V detector voltage, and 250 °C ion source.

## Primary metabolite data processing

Raw GC-TOF MS data files were preprocessed using ChromaTOF version 4.0 without smoothing, a 3 s peak width, baseline subtraction just above the noise level, and automatic mass spectral deconvolution and peak detection at signal/noise (s/n) levels of 5:1 throughout the chromatogram. Results were exported with absolute spectra intensities and further processed by a filtering algorithm implemented in the metabolomics BinBase database (*Skogerson et al., 2011*). The BinBase algorithm (rtx5) used the following settings: validity of chromatogram (107 counts/s), unbiased retention index marker detection (MS similarity >800, validity of intensity range for high m/z marker ions), retention index calculation by 5th order polynomial regression. Spectra were cut to 5% base peak abundance and matched to database entries from most to least abundant spectra using the following matching filters: retention index window ±2000 units (equivalent to about ±2 s retention time), validation of unique ions and apex masses (unique ion must be included in apexing masses and present at >3% of base peak abundance), mass spectrum similarity must fit criteria dependent on peak purity and signal/ noise ratios and a final isomer filter. Failed spectra were automatically entered as new database entries if signal/noise ratios were larger than 25 and mass spectral purity better than 80%. Data was reported as peak height using the unique quantification ion at the specific retention index, unless a different quantification ion was manually set in the BinBase administration software BinView.

## Lipid data acquisition

Lipid abundances were determined by charged-surface hybrid column-electrospray ionization quadrupole time-of-flight tandem mass spectrometry (CSH-ESI QTOF MS/MS). For positively charged lipids, an Agilent 6,530 QTOF mass spectrometer with resolution 10,000 was used and for negatively charged lipids, an Agilent 6,550 QTOF mass spectrometer with resolution 20,000 was used. Electrospray ionization was used to ionize column elutants in both positive and negative modes. Compounds were separated using a Waters Acquity ultra-high-pressure, liquid-chromatography charged surface hybrid column (UPLC CSH) C18 (100 mm length ×2.1 mm internal diameter; 1.7 µm particles). The conditions in positive mode were as follows: mobile phase A (60:40) acetonitrile:water +10 mM ammonium formiate +0.1% formic acid, mobile phase B (90:10) isopropanol:acetonitrile +10 mM ammonium formiate +0.1% formic acid. The conditions in negative mode were as follows: mobile phase A (60:40) acetonitrile:water +10 mM ammonium acetate, mobile phase B (90:10

isopropanol:acetonitrile +10 mM ammonium acetate). 5 µL of each brain, liver, and muscle sample was injected in negative mode. 0.5 µL of each brain and liver, and 0.25 µL of muscle samples was injected in positive mode. In both modes, the column temperature was 65 °C, at a flow rate of 0.6 mL/min, an injection temperature of 4 °C, and a gradient of 0 min 15%, 0–2 min 30%, 2–2.5 min 48%, 2.5–11 min 82%, 11–11.5 min 99%, 11.5–12 min 99%, 12–12.1 min 15%, and 12.1–15 min 15%. The ESI capillary voltage was set to +3.5 and –3.5 kV, and the collision energy to 25 for positive and negative modes. Mass-to-charge ratios (m/z) were scanned from 60 to 1200 Da and spectra acquired every 2 s. Automatic valve switching was used after each injection to reduce sample carryover for highly lipophilic compounds.

## Lipid data processing

Raw lipidomic data were processed using MS-DIAL *Tsugawa et al., 2015* followed by blank subtractions in Microsoft Excel and data cleanup using MS-FLO (*DeFelice et al., 2017*). Briefly, data were converted to files using Abf Converter. All default parameters were used for processing of MS-DIAL data, except for minimum peak height and width which were adjusted to the instrument. Results are exported from MS-DIAL and a blank reduction is performed for all features which are found in at least one sample. Blank reduction takes the maximum peak height relative to the blank average height and the average of all non-zero peak heights for samples. Duplicates and isotopes are examined using MS-FLO and deleted if confirmed. Peaks were annotated by manually comparing the MS/MS spectra and the accurate masses of precursor ions to spectra in the Fiehn laboratory LipidBlast spectral library (*Kind et al., 2013*). Additional peaks are manually curated from sample chromatograms. Manually curation was confirmed by using MassHunter Quant software to verify peak candidates based on peak shape and height reproducibility, and retention time reproducibility in replicate samples. The data were reported as peak heights for the specific quantification ion at the specific retention time.

## Quantification and statistical analysis

### 0.1.1 weight change and K-factor calculations

Percent weight change for each fish was calculated using formula 1. Mass and length measurements were recorded at the beginning and end of feeding regimens.

$$\Delta Wt(\%) = (m_{\text{final}} - m_{\text{initial}})/m_{\text{final}} \times 100, \tag{1}$$

where $m$ is mass. K-factor is a metric that represents both the mass and length of individuals and is frequently used in aquaculture research to assess an animal's physical condition (*Imam et al., 2010*). K-factor for each fish was calculated at the beginning of feeding regimens (app. 4 months) and on the day of dissection (30 days later) using the formula (c) below. Percent K-factor change was calculated using formula (d).

$$K = (m/x^3) \times 100, \tag{2}$$

where $x$ is the standard length.

$$\Delta K(\%) = (K_{\text{final}} - K_{\text{initial}})/K_{\text{final}} \times 100 \tag{3}$$

Data was first analyzed for normality using four independent methods: D'Agostino-Pearson, Shapiro-Wilk, Kolmogorov-Smirnov, Anderson-Darling. When comparing between more than two groups, data that passed three of four normality tests were analyzed using One-way ANOVA with Tukey correction for multiple comparisons between all groups. Data which failed more than one normality test, was analyzed with Kruskal-Wallis test using Dunn's for multiple comparison correction. The tests used in each figure are reported in the figure legends. p-values less than 0.05 are reported and the level of significance is indicated using the * system (ns, p > 0.05; *, $p \leq 0.05$; **, $p \leq 0.01$; ***, $p \leq 0.001$; ****, $p \leq 0.0001$).

## Further data processing

Processed primary metabolite data were vector normalized using mTIC. First, the sum of all peak heights for all identified metabolites, excluding the unknowns, for each sample was calculated. Such peak sums are called 'mTIC' and represent the sum of genuine metabolites (identified compounds) in

each sample. This method avoids unidentified peaks that could represent potentially non-biological artifacts such as column bleed, contaminants, or routine machine maintenance. mTIC averages for each sample were compared to determine if the variance between samples was significantly different ($p < 0.05$). Samples were then normalized to the average mTIC 'mTICaverage' within populations (Surface, Pachón, or Tinaja) and within organs (brain, muscle, or liver). For example, each biological replicate of the Tinaja brain group was normalized to the average mTIC of all Tinaja brain replicates regardless of feeding state. The equation (a) below was then used to normalize each metabolite (i) of a sample (j). After normalization, data are reported as 'relative semi-quantifications' or normalized peak heights.

$$y_{ij} \text{ (normalized)} = (x_{ij,raw}/\text{mTIC}_j) \times \overline{\text{mTIC}}, \tag{4}$$

where $x_{ij}$ is the raw peak intensity for metabolite in sample $j$, $\text{mTIC}_j$ is the average identified peak intensity in sample $j$, $\overline{\text{mTIC}}$ is the global average identified peak intensity, and $y_{ij}$ is the mTIC-normalized intensity of metabolite in sample $j$.

## Metabolite categorization

Metabolites were categorized according to their respective subclass classification in the human metabolite database (**Wishart et al., 2018**) (if the subclass was absent, we instead used the superclass of the respective metabolite). Metabolite classes with low membership were manually reassigned to arrive at five broad metabolite categories:

- Carbohydrates and central carbon metabolites (CCM). Simple sugars such as glucose, fructose, and various phosphates thereof, as well as core metabolites in glycolysis, gluconeogenesis, the TCA cycle, and the pentose phosphate pathway.
- Amino acids. All amino acids and intermediates in amino acid biosynthesis and degradation.
- Fatty acids. All free fatty acids, intermediates, and metabolites involved in lipogenesis and $\beta$-oxidation.
- Miscellaneous / secondary metabolites. Metabolites that do not fall in any of the other categories.
- Nucleotides. All nucleotides, nucleosides, nucleobases, and byproducts / intermediates of nucleotide metabolism.

Within each of metabolite category, we further normalized $log_{10}$ peak intensities using z-score normalization prior to performing PCA, O-PLS (described below) or any supervised classification or statistical modeling.

## O-PLS

In order to remove sources of variation not useful in discriminating the feeding state of different samples, we used O-PLS (**Trygg and Wold, 2002**), a technique commonly used in spectroscopy to correct for systematic variation (**Trygg and Wold, 2002**). O-PLS is often applied to raw spectra in order to eliminate the influence of background signals, but here we apply it instead to mTIC normalized peak intensities. Our main use of O-PLS is to remove biological noise that is uncorrelated with feeding state, such as baseline differences or trends among different populations. While z-score normalization already removes many of these artifacts, we observed that O-PLS generally enhanced the predictive accuracy of our PLS classifier. Given an input matrix $X$ of $n$ samples and $m$ spectral features (metabolite peak intensities in our study), and a target matrix $Y$ of classes or measured values (here, the feeding state), the final output of O-PLS (referred to here as $X'$) is again an $m$ by $n$ matrix consisting of $X$ with the systematic variation orthogonal to $Y$ removed.

## Characterization of feeding state responses

In order to determine which tissues and metabolite categories are most strongly implicated in **Aspiras et al., 2015** the starvation response within a given population, (**Xiong et al., 2018**) differences in metabolite levels between different populations for a given feeding state, we used a simple one-component PLS classifier trained on the output $X'$ of O-PLS.

The discriminant $Q^2$ value is a metric of PLS model accuracy and is given by

$$Q^2 = 1 - \frac{\sum_k (y_k - \hat{y}_k)^2}{\sum_k (y_k - \hat{y}_k)^2}$$

However, we use a truncated version $DQ^2$ (**Westerhuis et al., 2008**), where $y_k$ is replaced by $y'_k = \max(\min(y_k, 1), -1)$ and $y'_k$ is used in place of $y_k$. This metric does not penalize the PLS model for correct predictions that overshoot the target class label.

Using this framework, we employed a two-step model comprised of a O-PLS model followed by a single-component PLS model to discriminate refed versus long term-starved samples. We trained this combined model on z score-normalized log-transformed data for primary metabolites subdivided into categories. The output of the initial-stage O-PLS model consists of the original data with a PLS component representing 'orthogonal' noise removed. This de-noised data was then used to train a one-component PLS classifier on labels representing feeding state. This results in a $DQ^2$ value for the ability to discriminate refed versus starved states. Finally, an iterative scheme was used to randomly permute the label indices of the input data, resulting in a distribution of $DQ^2$ values. The significance level of the original predictive $DQ^2$ value was calculated using a two-tailed survival function of a normal distribution fitted to the $DQ^2$ values.

## Identification of significant metabolites

We employed a logistic regression model to identify important features (metabolites, lipids and classes thereof). We were specifically interested in marginal p-values of each individual metabolite, hence we constructed separate single-covariate models for each metabolite or lipid. Models were further based on different types of comparisons: (**Aspiras et al., 2015**) we compared different feeding states within a given population and (**Xiong et al., 2018**) different populations within a given feeding state. Logistic regression models (and GLMs in general) tend to suffer from complete separation of observed covariates (**Huang et al., 2020**). This renders maximum-likelihood estimates of the model parameters impossible. We therefore used the bayesglm function of the arm R package (**Gelman et al., 2008**) to obtain estimates for model coefficients, even in the case of complete separation. The bayesglm requires specifying a prior distribution. We found that the highly conservative default prior corresponding to a an assumption that the response to a change in input should typically not exceed roughly ±5 on the logistic scale, or, equivalently, no typical change in input should cause a shift in probability from 0.01 to 0.50, or 0.50–0.99 (**Gelman et al., 2008**) was sufficient to identify important metabolic changes in our comparisons. However, given the conservative nature of this prior distribution, we did not perform FDR correction.

We first split the input dataset into two matrices: one containing populations as category labels (bottom left), and one containing feeding states as category labels (upper right). We then subset each of these into the three possible pairwise comparisons from each group, compute z-score-normalized values within the comparison, filter the resulting matrix using O-PLS to remove orthogonal noise, and use the bayesglm function to fit a model to the respective comparison for discriminating either populations (within a given feeding state) or feeding states (within a given population). In each case, the GLM consists of a single covariant corresponding to metabolite / lipid peak heights (for individual metabolites) or classes of metabolites / lipids.

## Differences between feeding states and shared metabolites

To identify metabolites that might play a role in cave adaptation, we sought to fit logistic regression to an input capturing the difference between refed and starved samples and differences between cave and surface populations simultaneous. We implemented this using the following formula:

$$x = (P + T)/2 - S, \tag{5}$$

where $P$, $T$, and $S$ are z-score normalized mTIC peak intensities for starved vs. refed samples. In general, these vectors have length 12 (6 refed and 6 starved samples). A Bayesian logistic regression model was then fitted to the $x$ vector for each metabolite as before, with each element of the response vector labelled accordingly (starved or refed).

## ROS staining and microscopy

We repeated the original starvation experiment using 12 fish each from Pachòn and surface populations. The age of fish was 138 dpf at the time of collection. Following the original fasting procedure, six fish from each population were separated and fasted for 30 days. The remaining six fish in each population were fasted for 4 days and refed for at least three hours on the morning of collection. However, three fish from the surface 30 day fasted group did not survive until collection and were not used in the experiment.

Pachòn and surface fish were dissected alternately. Dissections were split across 2 days, with 3 fish each in all four experimental groups on the first day and 3 fish each in all groups except 30-day fasted surface on the second day. Livers were sectioned on 4% agarose gel and each section was stained with a primary dye and DAPI. Primary dyes included dihydroethidium (DHE), MitoSox, and MitoTracker. Sections were imaged with a ZEISS LSM 780 Laser Scanning Microscope. A Z-stack was taken at a location with suitable cell density for each sample.

Quantitation for cytoplasmic and nuclear intensity was accomplished by first taking a 2D slice from the middle plane. We created a threshold mask for the dye signal and DAPI using the average intensity of all Z-slices (not only the middle slice) and intensity measured on the middle plane to compute total intensity. We then took either the difference or intersection of the signal (DHE, MitoSox) and nuclear stain (DAPI) masks depending on whether cytoplasmic or nuclear signal was desired. We then integrated the signal and using the Kruskal-Wallis test to determine significance.

## Acknowledgements

JKM, JP, TB, LO, RP, JK, SX, and NR were supported by institutional funding from the Stowers Institute for Medical Research. Additionally, NR is supported by the Edward Mallinckrodt Foundation, NIH Grant R01 GM127872, DP2AG071466, NSF IOS-1933428, and EDGE award 1923372. This work was done to fulfill, in part, requirements for SX's PhD thesis research as a student registered with the Open University, UK. We are deeply grateful to Zachary Zakibe, Andrew Ingalls, Alba Aparicio Fernandez, David Jewell, and Franchesca Hutton-Lau, Molly Miller for their tireless efforts in maintaining the large fish fish population of the SIMR Cavefish Facility and providing invaluable assistance for this study and others. We would also like to thank Elizabeth Evans, Diana Baumann, M Shane Merryman, and the SIMR Reptile and Aquatics facility for coordinating and supporting animal research at Stowers and in the Rohner lab in particular, and Rhonda Egidy and Anoja Perera of the Stowers Molecular Biology Facility and Huzaifa Hassan of the Stowers Computational Biology Facility for library construction and analysis of the RNA-Seq study. We would also like to express our sincere thanks to Fengyan Deng, Hannah Wilson, Yongfu Wang, and Cathy McKinney of the Stowers Histology and Microscopy centers resp. for their help in sectioning, staining, and imaging experiments. We are grateful to the West Coast Metabolomics Center for providing data acquisition and analysis for this study.

## Additional information

### Funding

| Funder | Grant reference number | Author |
| --- | --- | --- |
| National Institutes of Health | 1DP2AG071466-01 | Nicolas Rohner |
| National Science Foundation | EDGE 1923372 | Nicolas Rohner |

The funders had no role in study design, data collection and interpretation, or the decision to submit the work for publication.

### Author contributions

J Kyle Medley, Conceptualization, Data curation, Formal analysis, Resources, Software, Validation, Visualization, Writing – original draft, Writing – review and editing; Jenna Persons, Conceptualization, Data curation, Formal analysis, Investigation, Visualization, Writing – review and editing; Tathagata

Biswas, Data curation, Resources; Luke Olsen, Methodology, Resources, Validation; Robert Peuß, Resources, Supervision; Jaya Krishnan, Resources, Supervision, Validation; Shaolei Xiong, Data curation, Investigation; Nicolas Rohner, Conceptualization, Funding acquisition, Project administration, Supervision, Visualization, Writing – review and editing

**Author ORCIDs**
Jenna Persons ⓘ http://orcid.org/0000-0002-8807-9678
Jaya Krishnan ⓘ http://orcid.org/0000-0003-2302-5748
Nicolas Rohner ⓘ http://orcid.org/0000-0003-3248-2772

**Ethics**
This study was performed in strict accordance with the recommendations in the Guide for the Care and Use of Laboratory Animals of the National Institutes of Health. All of the animals were handled according to approved institutional animal care and use committee (IACUC) protocols (#2021-129) of the Stowers Institute for Medical Research.

**Decision letter and Author response**
Decision letter https://doi.org/10.7554/eLife.74539.sa1
Author response https://doi.org/10.7554/eLife.74539.sa2

## Additional files

### Supplementary files
• Transparent reporting form

• Supplementary file 1. An Excel file containing the results of the O–PLS / GLM feature identification pipeline for primary metabolites divided into categories. Columns represent different population and feeding condition combinations.

• Supplementary file 2. GO term enrichment for the three RNA-Seq datasets described in this manuscript.

### Data availability
Source data and a complete pipeline are linked in the submission: http://www.stowers.org/research/publications/libpb-1699 (ftp://odr.stowers.org/LIBPB-1699), and has also been deposited to the github repository https://github.com/stowersinstitute/libpb-1699-metabolomics (copy archived at swh:1:rev:f4ac2098697e1d3e263dcaa72cc2761beb08bdba). An interactive Shiny application (https://cavefin.shinyapps.io/shiny/) can be used to explore and visualize our dataset.

The following dataset was generated:

| Author(s) | Year | Dataset title | Dataset URL | Database and Identifier |
|---|---|---|---|---|
| Kyle Medley J, Persons J, Biswas T, Olsen L, Peuß R, Krishnan J, Xiong S, Rohner N | 2022 | The Metabolome of Mexican Cavefish Shows a Convergent Signature Highlighting Sugar, Antioxidant, and Ageing-Related Metabolites | https://www.stowers.org/research/publications/libpb-1699 | Stowers Institute, LIBPB-1699 |

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
