## [Editor Report]

Medley et al., study *A. mexicanus*, an extreme-adapted organism with important connections to human health. The authors test metabolic responses in this natural model of elevated blood glucose and extensive body fat deposits, conditions generally expected to predispose to a higher risk for metabolic syndrome and higher frailty. The work is rigorous and will provide an important reference for future studies aimed at dissecting the mechanistic basis underlying metabolic shifts in this uniquely attractive model. The authors also provide an open and accessible window into their data and analyses by sharing a Shiny app.

---

## [Decision Letter]

**Decision letter after peer review:**

Thank you for submitting your article "Resilience in a Natural Model of Metabolic Dysfunction Through Changes in Longevity and Ageing-Related Metabolites" for consideration by *eLife*. Your article has been reviewed by 3 peer reviewers, including Dario Riccardo Valenzano as Reviewing Editor and Reviewer #1 and the evaluation has been overseen by Carlos Isales as the Senior Editor.

Essential revisions:

This work lays out a series of plausible associations between metabolic states in cave populations – based on metabolomics analyses of different organs – and cavefish biology.

The conclusions of this paper are based almost exclusively on metabolomics data. The reviewers feel that it would be very important to validate these results with alternative techniques (ex. transcriptomic analysis).

The title is misleading. It's not entirely clear whether the metabolomic analysis informs us into how metabolic resilience is achieved in cavefish. Hence, the word "resilience" should be removed from the title. Also, to be factual about in which species these results were obtained, the name "cavefish" (or Mexican tetra) should be featured in the title.

We would like to request major changes to this manuscript, in order to be further considered for publication in *eLife*.

The most relevant points are:

1. An independent validation of the metabolomics results (see reviewers comments).

2. An explanation about impact of fish age or include age as a potential confounder for the analyses. At least the authors should raise caveats about age differences among groups.

3. Remove Figure 2 or provide a clearer representation of their results.

4. Tone down the mechanistic relevance of their findings.

5. Fix/address inconsistency of Figure 5 (fructose upregulation in the brain).

6. Perform/report about multiple testing correction – provide a power calculation for the number of fish/tissues used to mitigate for the lack of experimental validation.

7. Openly address their vision in terms of physiological adaptations/positive selection vs. alternative evolutionary paths that might have led to cavefish specific phenotypic spectrum (including metabolism).

8. All reviewers feel the paper should be re-written as a resource paper.

*Reviewer #1 (Recommendations for the authors):*

The paper is well-written, but there are important points I'd like to bring up, which can strengthen it. It is clearly a resource paper, rather than a "finding-strong" work. This should be made more explicit.

The title is misleading. I am not entirely convinced this metabolomic analysis informs us into how metabolic resilience is achieved in cavefish. I suggest removing the word "Resilience" from the title. Also, I suggest the name "cavefish" should be featured in the title.

There is no attempt to go beyond association between metabolic states and functional metabolic "adaptations" (which is OK, descriptive work is important).

It is not clear what these adaptations the authors speak about are exactly. Are they adaptations to lack of nutrients, light, different temperature in the cave environment, all of the above?

I find this work is generally well conducted; however, often the experimental approaches are presented as results (e.g. Figure 3). The authors should separate better the methods section from the results part. The first paragraph of the Results section does not necessarily belongs to the results part. It is about "what was done", rather than "here is question A, here is our approach, here is the result, here is our comment/interpretation of the result".

Figure 3 has basically no clear message, other than "what was done" and how the samples separate. I figure the message is "cavefish populations clusters closer to one another than to the surface population based on a, b and c. Perhaps this should be stated declaratively in the caption.

In Figure 2 there is a possible typo: "Experimental setup (A) and PCA…". I do not see any PCA, if this refers to Principal Component Analysis. Perhaps this caption is the relict of an older version of the figure? Please amend.

Generally, I am un-easy with the use of the phrase "evolved populations". Also surface populations are evolved. Perhaps "recently derived" would be more apt?

*Reviewer #2 (Recommendations for the authors):*

As a result of using metabolomics without further validation or further going into at least one aspect of the findings in much more detail using other techniques, the manuscript lacks impact and the conclusions drawn are very speculative. At the very least I would like to see the main metabolites identified to be different validated using another method to be able to confidently draw conclusions. But to increase the impact of the results for the field, I would ideally like to see further experiments to investigate one of the main findings in detail.

Have power calculations been used to determine sample size? an n of 6 seems low for metabolomics with the observed variations to be able to draw the conclusions drawn. This needs to be shown and sample size needs to be justified/increased. This is extra relevant because of the lack of validation.

---

## [Author Response]

Essential revisions:This work lays out a series of plausible associations between metabolic states in cave populations – based on metabolomics analyses of different organs – and cavefish biology.The conclusions of this paper are based almost exclusively on metabolomics data. The reviewers feel that it would be very important to validate these results with alternative techniques (ex. transcriptomic analysis).

We have included the results of several RNA-Seq experiments (including an age-matched dataset) in the revised submission that we hope will address this gap. Please see (1) below for more information.

The title is misleading. It's not entirely clear whether the metabolomic analysis informs us into how metabolic resilience is achieved in cavefish. Hence, the word "resilience" should be removed from the title. Also, to be factual about in which species these results were obtained, the name "cavefish" (or Mexican tetra) should be featured in the title.

We agree that the original title was not optimal and have incorporated this suggestion in the revised submission.

We would like to request major changes to this manuscript, in order to be further considered for publication in eLife.The most relevant points are:1. An independent validation of the metabolomics results (see reviewers comments).

In order to address this point, we have added Supplementary Figures 1 and 2, which include three RNA-Seq datasets (one of which is age-matched to this study). The transcriptomics data confirms the main themes of the metabolomics data – alterations of sugar and antioxidant metabolism. We discuss in the text how these trends vary between the different transcriptomics datasets. Steady-state levels of metabolites are, of course, a function of many variables, and hence RNA-Seq data can only be used for qualitative comparisons (barring building a complex model), but we find the sugar and antioxidant metabolism gene signatures encouraging.

2. An explanation about impact of fish age or include age as a potential confounder for the analyses. At least the authors should raise caveats about age differences among groups.

The new data added in point (1) above includes an RNA-Seq dataset that is exactly age-matched with the fish in this study, which helps address this point. Furthermore, as part of our resubmission, we have removed data that we feel to be extraneous and does not contribute to the theme of the main text (a metabolomic survey of surface and cave populations of *A. mexicanus*), and as a result some of the non-age matched results have also been removed. We have noted age differences where still applicable.

3. Remove Figure 2 or provide a clearer representation of their results.

We agree that the Venn diagrams in the original manuscript (we believe the reviewer was actually referring to Figure 4) were confusing and not very lucid, and so we have removed the figure.

4. Tone down the mechanistic relevance of their findings.

We appreciate the reviewers’ objective assessment of the findings. Upon consideration of the reviewer comments, we agree that the findings presented here to not meet the standards required to draw mechanistic conclusions and have attempted to remove mechanistic language and speculative comparisons from the manuscript. We hope future studies will establish connections between the these metabolic trends and mechanistic details of cave adaptation. We believe establishing *A. mexicanus* cell lines will help elucidate these mechanisms (cell lines are particularly useful because of the availability of gene editing tools, whereas gene editing in fish embryos is technically challenging). This is an effort that our group is actively engaged in but will take some time to achieve.

5. Fix/address inconsistency of Figure 5 (fructose upregulation in the brain).

We regret any confusion caused by Figure 5 (now Figure 4) and have renormalized it on a per-tissue basis to show the trend in brain fructose levels. The figure now clearly shows that fructose levels are higher in cave populations. As an aside, the fructose comment was a comparison against naked mole-rats. As several reviewers observed, these comparisons are speculative, and we removed them for this reason (hence the original comment related to fructose no longer exists in the current revision).

6. Perform/report about multiple testing correction – provide a power calculation for the number of fish/tissues used to mitigate for the lack of experimental validation.

We acknowledge the concerns raised by reviewers regarding the sample size below. Briefly, we are aware of the difficulty in drawing statistical conclusions for the sample configuration used in our study. We chose to split our budget between primary metabolites and lipids, and even so the combined dataset was considerably expensive to obtain (~$50k USD).

If we had instead chosen to focus on either primary metabolites or lipids, we could have employed a sample size of n=12, which is more statistically sound. However, given the exploratory nature of this work and the fact that this represents the first metabolomics study of *A. mexicanus* cavefish, we believe that a more comprehensive dataset is more useful than a smaller, but more statistically powerful dataset. We hope this manuscript will help guide future hypothesis-based mechanistic studies, rather than serve to confirm or reject a hypothesis based on statistics.

As for multiple testing correction, it is usually not applied to Bayesian analyses such as ours. We chose a Bayesian approach because we were unable to fit a frequentist GLM to our data (due to “perfect separation”). This confirms that sample size is an issue.

However, in Bayesian analyses, the choice of prior distribution can be thought of as a proxy for “false positives” (although false positives are a frequentist concept). In other words, reviewers who are more familiar with frequentist statistics can think of the choice of prior as a substitute for applying false discovery rate correction, and we hope our argument will convince the reviewers that our choice of prior can be used on lieu of a multiple testing correction.

The prior used by our analysis is highly conservative (spread across roughly ±5 of the logistic scale, see Gelman 2008 reference). Thus, reported statistics can be thought of as being a conservative estimate. A power analysis of a Bayesian GLM is beyond our technical abilities.

Finally, we hope this point is also addressed by our inclusion of multiple sets of RNA-Seq data (including age-matched data).

7. Openly address their vision in terms of physiological adaptations/positive selection vs. alternative evolutionary paths that might have led to cavefish specific phenotypic spectrum (including metabolism).

We have attempted to clean up the manuscript and removing figures / data that did not support the main theme of metabolic characterization of *A. mexicanus* cave and surface fish, and we have also followed reviewer suggestions about removing references to positive selection. This cleaned-up version will hopefully offer a clearer picture of our main work.

8. All reviewers feel the paper should be re-written as a resource paper.

We appreciate the suggestion to resubmit the manuscript as a resource paper and agree that this article type is more appropriate. Upon investigation of formatting requirements for resource papers, we discovered that the current layout of the manuscript already meets the requirements of a resource paper, and the only additional requirement is to publish all source code under the OSI open-source definition. As with the initial submission, we have included a link for the reviewers to download the source code and will publish it publicly under an OSI-approved license pending acceptance of the manuscript.

Reviewer #1 (Recommendations for the authors):The paper is well-written, but there are important points I'd like to bring up, which can strengthen it. It is clearly a resource paper, rather than a "finding-strong" work. This should be made more explicit.

We are very grateful for the reviewer’s suggestions. We have ensured the manuscript is compliant with *eLife* resource paper guidelines and toned down the mechanistic language.

The title is misleading. I am not entirely convinced this metabolomic analysis informs us into how metabolic resilience is achieved in cavefish. I suggest removing the word "Resilience" from the title. Also, I suggest the name "cavefish" should be featured in the title.

We appreciate these suggestions and agree the title can be improved. The revised manuscript incorporates these suggestions.

There is no attempt to go beyond association between metabolic states and functional metabolic "adaptations" (which is OK, descriptive work is important).It is not clear what these adaptations the authors speak about are exactly. Are they adaptations to lack of nutrients, light, different temperature in the cave environment, all of the above?

While there are a number of adaptations in cavefish to all of the environmental challenges the reviewer mentioned, we believe the findings here most relate to (1) extreme nutrient scarcity (to be more explicit, long periods of nutrient scarcity interrupted by seasonal flooding) and (2) secondary conditions triggered by this (e.g., excess fat accumulation, which is normally detrimental). We believe changes in sugar metabolism are related to the former whereas changes in antioxidant metabolism are related to the latter. We have endeavored to better emphasize this point in the text, particularly in the third paragraph of the introduction.

I find this work is generally well conducted; however, often the experimental approaches are presented as results (e.g. Figure 3). The authors should separate better the methods section from the results part. The first paragraph of the Results section does not necessarily belongs to the results part. It is about "what was done", rather than "here is question A, here is our approach, here is the result, here is our comment/interpretation of the result".

Upon re-reading this paragraph we agree with the reviewer that it is out-of-place in the Results section, and we have moved it to the introduction.

Figure 3 has basically no clear message, other than "what was done" and how the samples separate. I figure the message is "cavefish populations clusters closer to one another than to the surface population based on a, b and c. Perhaps this should be stated declaratively in the caption.

We agree that the message of this figure can be more clearly stated. We have incorporated the reviewer’s suggestion and revised the figure caption to point out the sample association patterns.

In Figure 2 there is a possible typo: "Experimental setup (A) and PCA…". I do not see any PCA, if this refers to Principal Component Analysis. Perhaps this caption is the relict of an older version of the figure? Please amend.

We are very grateful to the reviewer for catching this mistake! This did indeed refer to an earlier version of the manuscript. We have corrected the typo.

Generally, I am un-easy with the use of the phrase "evolved populations". Also surface populations are evolved. Perhaps "recently derived" would be more apt?

The reviewer’s point is well-taken, particularly in a metabolomic study which with no genetic component. We have made the suggested corrections to the manuscript and replaced all occurrences of “evolved populations.”

Reviewer #2 (Recommendations for the authors):As a result of using metabolomics without further validation or further going into at least one aspect of the findings in much more detail using other techniques, the manuscript lacks impact and the conclusions drawn are very speculative. At the very least I would like to see the main metabolites identified to be different validated using another method to be able to confidently draw conclusions. But to increase the impact of the results for the field, I would ideally like to see further experiments to investigate one of the main findings in detail.

We agree with the reviewer on the need for additional validation. To this end, we have included several RNA-Seq datasets in Figure S1/2, including an age-matched dataset. While it is not possible to draw quantitative conclusions about the steady state levels of metabolites from alterations in gene expression (without a complex model), the main themes of our manuscript – alterations of antioxidant and sugar metabolites – are also represented in the transcriptional data.

Have power calculations been used to determine sample size? an n of 6 seems low for metabolomics with the observed variations to be able to draw the conclusions drawn. This needs to be shown and sample size needs to be justified/increased. This is extra relevant because of the lack of validation.

The reviewer is correct that n=6 is probably too low. We made the decision to use a sub-optimal sample size in order to cover both primary metabolites and lipids, and even so the data acquired for this study was considerably expensive (~50k USD). If we had instead covered only one of the two, we could have used n=12, but given the exploratory nature of this work we elected to cover more metabolites at the expense of drawing statistical conclusions. The statistical model we use (a Bayesian GLM) is rather complex and a power analysis of it is beyond our technical abilities (and the bayesglm function in the “arm” R package that we use does not appear to be designed for this). However, the model is based on a highly conservative prior and hence the estimates it gives can generally be trusted as a lower bound for effect size.